# PROBABILISTIC NUMERIC CONVOLUTIONAL NEURAL NETWORKS

**Marc Finzi**[*]
Qualcomm AI Research
New York University
maf820@nyu.edu

**Roberto Bondesan & Max Welling**
Qualcomm AI Research[†]
{rbondesa, mwelling}@qti.qualcomm.com

## ABSTRACT

Continuous input signals like images and time series that are irregularly sampled or have missing values are challenging for existing deep learning methods. Coherently defined feature representations must depend on the values in unobserved regions of the input. Drawing from the work in probabilistic numerics, we propose Probabilistic Numeric Convolutional Neural Networks which represent features as Gaussian processes (GPs), providing a probabilistic description of discretization error. We then define a convolutional layer as the evolution of a PDE defined on this GP, followed by a nonlinearity. This approach also naturally admits steerable equivariant convolutions under e.g. the rotation group. In experiments we show that our approach yields a $3\times$ reduction of error from the previous state of the art on the SuperPixel-MNIST dataset and competitive performance on the medical time series dataset PhysioNet2012.

## 1 INTRODUCTION

Standard convolutional neural networks are defined on a regular input grid. For continuous signals like time series and images, these elements correspond to regular samples of an underlying function $f$ defined on a continuous domain. In this case, the standard convolutional layer of a neural network is a numerical approximation of a continuous convolution operator $\mathcal{A}$.

Coherently defined networks on continuous functions should *only* depend on the input function $f$, and not on spurious shortcut features (Geirhos et al., 2020) such as the sampling locations or sampling density, which enable overfitting and reduce robustness to changes in the sampling procedure. Each application of $\mathcal{A}$ in a standard neural network incurs some discretization error which is determined by the sampling resolution. In some sense, this error is unavoidable because the features $f^{(\ell)}$ at the layers $\ell$ depend on the values of the input function $f$ at regions that *have not been observed*. For input signals which are sampled at a low resolution, or even sampled irregularly such as with the sporadic measurements of patient vitals data in ICUs or dispersed sensors for measuring ocean currents, this discretization error cannot be neglected. Simply filling in the missing data with zeros or imputing the values is not sufficient since many different imputations are possible, each of which can affect the outcomes of the network.

Probabilistic numerics is an emergent field that studies discretization errors in numerical algorithms using probability theory (Cockayne et al., 2019). Here we build upon these ideas to quantify the dependence of the network on the regions in the input which are unknown, and integrate this uncertainty into the computation of the network. To do so, we replace the discretely evaluated feature maps $\{f^{(\ell)}(x_i)\}_{i=1}^N$ with Gaussian processes: distributions over the continuous function $f^{(\ell)}$ that track the most likely values as well as the uncertainty. On this Gaussian process feature representation, we need not resort to discretizing the convolution operator $\mathcal{A}$ as in a standard convnet, but instead we can apply the continuous convolution operator directly. If a given feature is a Gaussian process, then applying linear operators yields a new Gaussian process with transformed mean and covariance functions. The dependence of $\mathcal{A}f$ on regions of $f$ which are not known translates into

---

[*]Work done during internship at Qualcomm AI Research
[†]Qualcomm AI Research is an initiative of Qualcomm Technologies, Inc.

the uncertainty represented in the transformed covariance function, the analogue of the discretization error in a CNN, which is now tracked explicitly. We call the resulting model Probabilistic Numeric Convolutional Neural Network (PNCNN).

## 2   RELATED WORK

Over the years there have been many successful convolutional approaches for ungridded data such as GCN (Kipf and Welling, 2016), PointNet (Qi et al., 2017), Transformer (Vaswani et al., 2017), Deep Sets (Zaheer et al., 2017), SplineCNN (Fey et al., 2018), PCNN (Atzmon et al., 2018), PointConv (Wu et al., 2019), KPConv (Thomas et al., 2019) and many others (de Haan et al., 2020; Finzi et al., 2020; Schütt et al., 2017; Wang et al., 2018). However, the target domains of sets, graphs, and point clouds are intrinsically discrete and for continuous data each of these methods fail to take full advantage of the assumption that the underlying signal is continuous. Furthermore, none of these approaches reason about the underlying signal probabilistically.

In a separate line of work there are several approaches tackling irregularly spaced time series with RNNs (Che et al., 2018), Neural ODEs (Rubanova et al., 2019), imputation to a regular grid (Li and Marlin, 2016; Futoma et al., 2017; Shukla and Marlin, 2019; Fortuin et al., 2020), set functions (Horn et al., 2019) and attention (Shukla and Marlin, 2020). Additionally there are several works exploring reconstruction of images from incomplete observations for downstream classification (Huijben et al., 2019; Li and Marlin, 2020).

Most similar to our method are the end-to-end Gaussian process adapter (Li and Marlin, 2016) and the multi-task Gaussian process RNN classifier (Futoma et al., 2017). In these two works, a Gaussian process is fit to an irregularly spaced time series and sampled imputations from this process are fed into a separate RNN classifier. Unlike our approach where the classifier operates directly on a continuous and probabilistic signal, in these works the classifier operates on a deterministic signal on a regular grid and cannot reason probabilistically about discretization errors.

Finally, while superficially similar to Deep GPs (Damianou and Lawrence, 2013) or Deep Differential Gaussian Process Flows (Hegde et al., 2018), our PNCNNs tackle fundamentally different kinds of problems like image classification[1], and our GPs represent epistemic uncertainty over the values of the feature maps rather than the parameters of the network.

## 3   BACKGROUND

**Probabilistic Numerics:**    We draw inspiration for our approach from the community of *probabilistic numerics* where the error in numerical algorithms are modeled probabilistically, and typically with a Gaussian process. In this framework, only a finite number of input function calls can be made, and therefore the numerical algorithm can be viewed as an autonomous agent which has epistemic uncertainty over the values of the input. A well known example is Bayesian Monte Carlo where a Gaussian process is used to model the error in the numerical estimation of an integral and optimally select a rule for its computation (Minka, 2000; Rasmussen and Ghahramani, 2003). Probabilistic numerics has been applied widely to numerical problems such as the inversion of a matrix (Hennig, 2015), the solution of an ODE (Schober et al., 2019), a meshless solution to boundary value PDEs (Cockayne et al., 2016), and other numerical problems (Cockayne et al., 2019). To our knowledge, we are the first to construct a probabilistic numeric method for convolutional neural networks.

**Gaussian Processes:**    We are interested in operating on the continuous function $f(x)$ underlying the input, but in practice we have access only to a collection of the values of that function sampled on a finite number of points $\{x_i\}_{i=1}^N$. Classical interpolation theory reconstructs $f$ deterministically by assuming a certain structure of the signal in the frequency domain. Gaussian processes give a way of modeling our beliefs about values that have not been observed (Rasmussen et al., 2006), as reviewed in appendix A. These beliefs are encoded into a prior covariance $k$ of the GP $f \sim \mathcal{GP}(0, k)$ and updated upon seeing data with Bayesian inference. Explicitly, given a set of sampling locations $\boldsymbol{x} = \{x_i\}_{i=1}^N$ and noisy observations $\boldsymbol{y} = \{y_i\}_{i=1}^N$ sampled $y_i \sim \mathcal{N}(f(x_i), \sigma_i^2)$, using Bayes rule

---

[1]While GPs *can* be applied directly to image classification, they are not well suited to this task even with convolutional structure baked in, as shown in Kumar et al. (2018).

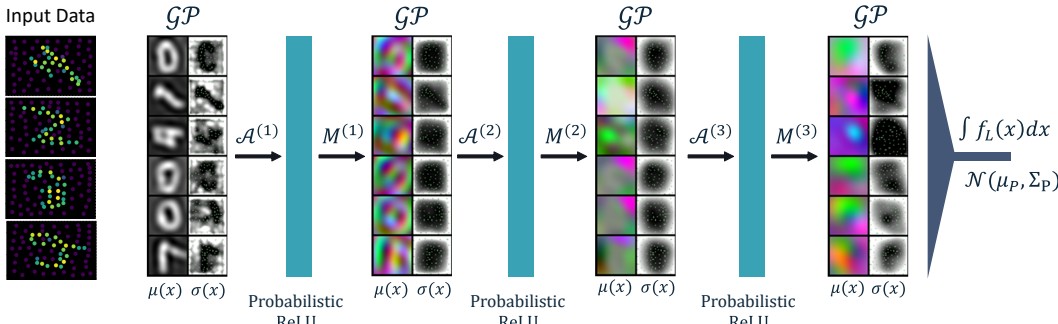

Figure 1: The PNCNN operating on SuperPixel-MNIST images shown on the left. The mean and elementwise uncertainty of the Gaussian process feature maps are shown as they are transformed through the network by the convolution layers. Observation points shown as green dots in $\sigma(x)$.

one can compute the posterior distribution $f|\boldsymbol{y}, \boldsymbol{x} \sim \mathcal{GP}(\mu_p, k_p)$, which captures our epistemic uncertainty about the values between observations. The posterior mean and covariance are given by

$$\mu_p(x) = \boldsymbol{k}(x)^\top [K + S]^{-1} \boldsymbol{y}, \quad k_p(x, x') = k(x, x') - \boldsymbol{k}(x)^\top [K + S]^{-1} \boldsymbol{k}(x'), \tag{1}$$

where $K_{ij} = k(x_i, x_j), k(x)_i = k(x, x_i)$ and $S = \mathrm{diag}(\sigma_i^2)$. Below we shall choose the RBF kernel[2] as prior covariance, due to its convenient analytical properties: $k_{\mathrm{RBF}}(x, x') = a\mathcal{N}(x; x', l^2 I) = a\left(2\pi l^2\right)^{-\frac{d}{2}} \exp(-\frac{1}{2l^2}||x - x'||^2)$. In typical applications of GPs to machine learning tasks such as regression, the function $f$ that we want to predict is already the regression model. In contrast, here we use GPs as a way of representing our beliefs and epistemic uncertainty about the values of both the input function and the intermediate feature maps of a neural network.

## 4    PROBABILISTIC NUMERIC CONVOLUTIONAL NEURAL NETWORKS

### 4.1    OVERVIEW

Given an input signal $f : \mathcal{X} \to \mathbb{R}^c$, we define a network with layers that act directly on this continuous input signal. We define our neural network recursively from the input $f^{(0)} = f$, as a series of $L$ continuous convolutions $\mathcal{A}^{(\ell)}$ with pointwise ReLU nonlinearities and weight matrices which mix only channels (known as $1 \times 1$ convolutions) $M \in \mathbb{R}^{c \times c}$:

$$f^{(\ell+1)} = M^{(\ell)}\mathrm{ReLU}[\mathcal{A}^{(\ell)} f^{(\ell)}], \tag{2}$$

and a final global average pooling layer $\mathcal{P}$ which acts channel-wise as natural generalization of the discrete case: $\mathcal{P}(f^{(L)})_\alpha = \int f_\alpha^{(L)}(x)\mathrm{d}x$ for each $\alpha = 1, 2, \ldots, c$. Denoting the space of functions on $\mathcal{X}$ with $c$ channels by $\mathcal{H}_c$, the convolution operators $\mathcal{A}^{(\ell)}$ are linear operators from $\mathcal{H}_{c_\ell}$ to $\mathcal{H}_{c_{\ell+1}}$. Like in ordinary convolutional neural networks, the layers build up increasingly more expressive spatial features and depend on the parameters in $\mathcal{A}^{(\ell)}$ and $M^{(\ell)}$. Unlike ordinary convolutional networks, these layers are well defined operations on the underlying continuous signal.

While it is clear that such a network can be defined abstractly, the exact values of the function $f^{(L)}$ cannot be computed as the operators depend on unknown values of the input. However, by adapting a probabilistic description we can formulate our ignorance of $f^{(0)}$ with a Gaussian process and see how the uncertainties propagate through the layers of the network, yielding a probabilistic output. Before delving into details, we outline the key components of equation 2 that make this possible.

**Continuous Convolutional Layers:** Crucially, we consider continuous convolution operators $\mathcal{A}$ that can be applied to input Gaussian process $f \sim \mathcal{GP}(\mu_p, k_p)$ in closed form. The output is another Gaussian process with a transformed mean and covariance $\mathcal{A}f \sim \mathcal{GP}(\mathcal{A}\mu_p, \mathcal{A}k_p\mathcal{A}')$ where $\mathcal{A}'$ acts

---

[2]For convenience, we include the additional scale factor $\left(2\pi l^2\right)^{d/2}$ relative to the usual definition.

to the left on the primed argument of $k_p(x, x')$.[3] In section 4.2 we show how to parametrize these continuous convolutions in terms of the flow of a PDE and show how they can be applied to the RBF kernel exactly in closed form.

**Probabilistic ReLUs:** Applying the ReLU nonlinearity to the GP yields a new non Gaussian stochastic process $h^{(\ell)} = \text{ReLU}[\mathcal{A}^{(\ell)} f^{(\ell)}]$, and we show in section 4.5 that the mean and covariance of this process has a closed form solution which can be computed.

**Channel Mixing and Central Limit Theorem:** The activations $h^{(\ell)}$ are not Gaussian; however, for a large number of weakly dependent channels we argue that $f^{(\ell+1)} = M^{(\ell)} h^{(\ell)}$ is approximately distributed as a Gaussian Process in section 4.5.

**Measurement and Projection to RBF Gaussian Process:** While $f^{(\ell+1)}$ is approximately a Gaussian process, the mean and covariance functions have a complicated form. Instead of using these functions directly, we take measurements of the mean and variance of this process and feed them in as noisy observations to a fresh RBF kernel GP, allowing us to repeat the process and build up multiple layers without increasing complexity.

The Gaussian process feature maps in the final layer $f^{(L)}$ are aggregated spatially by the integral pooling $\mathcal{P}$ which can also be applied in closed form (see appendix D), to yield a Gaussian output. Assembling these components, we implement the end to end trainable Probabilistic Numeric Convolutional Neural Network which integrates a probabilistic description of missing data and discretization error inherent to continuous signals. The layers of the network are shown in figure 1.

## 4.2 CONTINUOUS CONVOLUTIONAL LAYERS

On a discrete domain such as the lattice $\mathcal{X} = \mathbb{Z}^d$, all translation equivariant linear operators $\mathcal{A}$ are convolutions, a fact which we review in appendix B. In general, these convolutions can be written in terms of a linear combination of powers of the generators of the translation group: the shift operators $\tau_i, i = 1, \ldots, d$ shift all elements by one unit along the $i$-th axis of the grid. For a one dimensional grid, one can always write $\mathcal{A} = \sum_k W_k \tau^k$ where the weight matrices $W_k \in \mathbb{R}^{c \times c}$ act only on the channels and the shift operator $\tau$ acts on functions on the lattice. In $d$ dimensions, $\mathcal{A} = \sum_{k_1, \ldots, k_d} W_{k_1, \ldots, k_d} \tau_1^{k_1} \cdots \tau_d^{k_d}$ for some set of integer coefficients $k_1, \ldots, k_d$. For example when $d = 2$, we can take $k_1, k_2 \in \{-1, 0, 1\}$ to fill out a $3 \times 3$ neighborhood.

On the continuous domain $\mathcal{X} = \mathbb{R}^d$ we similarly parametrize convolutions with $\mathcal{A} = \sum_k W_k e^{\mathcal{D}_k}$, where $\mathcal{D}_k$ is given by powers of the partial derivatives $\partial_i, i = 1, \ldots, d$ which generate infinitesimal translations along the $i$-th axes. Setting $d = 1$ for simplicity, we can indeed verify that the operator exponential $\tau^a = e^{a \partial_x}$ applied to a function $g(x)$ is a translation: $e^{a \partial_x} g(x) = g(x) + a g'(x) + \frac{1}{2} a^2 g''(x) + \cdots = g(x+a)$, which is the Taylor series expansion of $g(x+a)$ around $x$. Exponentials of operators can be defined similarly in terms of the formal Taylor series $e^{\mathcal{D}} = \sum_{k=0}^{\infty} \mathcal{D}^k / k!$ or more broadly as the solution to the PDE:

$$\partial_t g(t, x) = (\mathcal{D}g)(t, x), \quad g(0, x) = g(x) \tag{3}$$

at time $t = 1$: $e^{\mathcal{D}} g(x) = g(t = 1, x)$.

Following the discussion in the discrete case, translation invariance of $\mathcal{D}_k$ imposes that it is expressed in terms of powers of the generators. Collecting the derivatives into the gradient $\nabla$, we can write the general form of $\mathcal{D}_k$ as $\alpha_k + \beta_k^\top \nabla + \frac{1}{2} \nabla^\top \Sigma_k \nabla + \ldots$ for any constants $\alpha_k$, vectors $\beta_k$, matrices $\Sigma_k$ etc. For simplicity, we truncate the series at second order to get

$$\mathcal{D}_k = \beta_k^\top \nabla + \tfrac{1}{2} \nabla^\top \Sigma_k \nabla, \tag{4}$$

where we omit the constants $\alpha_k$ that can be absorbed into the definition of $W_k$. For this choice of $\mathcal{D}$, the PDE in equation 3 is nothing but the diffusion equation with drift $\beta_k$ and diffusion $\Sigma_k$. When discussing rotational equivariance in section 4.4, we also consider a more general form of $\mathcal{D}$.

The diffusion layer can also be viewed in another way as the infinitesimal generator of an Ito diffusion (a stochastic process). Given an Ito process with constant drift and diffusion $dX_t =$

---

[3]More generally neural networks have affine layers including both convolutions and biases. An affine transformation $\mathcal{A}f + b$ of a Gaussian process is also a Gaussian process $f \sim \mathcal{GP}(\mathcal{A}\mu_p + b, \mathcal{A}k_p \mathcal{A}')$, and we include biases in our network but omit them from the derivations for simplicity.

$\beta \mathrm{d}t + \Sigma^{1/2}\mathrm{d}B_t$ where $B_t$ is a $d$ dimensional Brownian motion, the time evolution operator can be written via the Feynman-Kac formula as $\mathrm{e}^{t\mathcal{D}}f(x) = \mathbb{E}[f(X_t)]$ where $X_0 = x$. In other words, the operator layer $\mathcal{A} = \mathrm{e}^{t\mathcal{D}}$ is the expectation under a parametrized Neural Stochastic Differential equation (Li et al., 2020; Tzen and Raginsky, 2019) that is homogeneous and therefore shift invariant. The flow of this SDE depends on the drift and diffusion parameters $\beta$ and $\Sigma$.

To recap, we define our convolution operator through the general form $\mathcal{A} = \sum_k W_k \mathrm{e}^{\mathcal{D}_k}$ where the weight matrices $W_k \in \mathbb{R}^{c \times c}$ mix only channels and $\mathrm{e}^{\mathcal{D}_k}$ is the forward evolution by one unit of time of the diffusion equation with drift $\beta_k$ and diffusion $\Sigma_k$ containing learnable parameters $\{(W_k, \beta_k, \Sigma_k)\}_{k=1}^K$. The translation equivariance of $\mathcal{A}$ follows directly from the fact that the generators commute $\forall k, i : [\mathcal{D}_k, \nabla_i] = 0$ and therefore $[\mathcal{A}, \tau_i] = 0$ (the bracket $[a, b] = ab - ba$ is the commutator of the two operators). In appendix B we show that our definition of $\mathcal{A}$ reduces to the usual one in the discrete case and is thus a principled generalization to the continuous domain.

### 4.3 EXACT APPLICATION ON RBF GPS

Although the application of the linear operator $\mathcal{A} = \sum_k W_k \mathrm{e}^{\mathcal{D}_k}$ involves the time evolution of a PDE, owing to properties of the RBF kernel we fortuitously can apply the operator to an input GP in closed form! Gaussian processes are closed under linear transformations: given $f \sim \mathcal{GP}(\mu_p, k_p)$, we need only compute the action of $\mathcal{A}$ on the mean and covariance: $\mathcal{A}f \sim \mathcal{GP}(\mathcal{A}\mu_p, \mathcal{A}k_p\mathcal{A}')$, where $\mathcal{A}'$ is the adjoint w.r.t. the $L_2(\mathcal{X})$ inner product. The application of time evolution $\mathrm{e}^{\mathcal{D}_k}$ is a convolution with a Green's function $G_k$, so $\mathcal{A}f = \sum_k W_k \mathrm{e}^{\mathcal{D}_k}f = \sum_k W_k G_k * f$. As we derive in appendix C, the Green's function for $\mathcal{D}_k = \beta_k^\top \nabla + (1/2)\nabla^\top \Sigma_k \nabla$, is nothing but the multivariate Gaussian density $G_k(x) = \mathcal{N}(x; -\beta_k, \Sigma_k)$:

$$\mathcal{A}f = \sum_k W_k \mathrm{e}^{\mathcal{D}_k}f = \sum_k W_k G_k * f = \sum_k W_k \mathcal{N}(-\beta_k, \Sigma_k) * f. \tag{5}$$

In order to apply $\mathrm{e}^{t\mathcal{D}}$ to the posterior GP, we need only to be able to apply the operator to the posterior mean and covariance. This posterior mean and covariance in equation 1 are expressed in terms of $k_{\mathrm{RBF}} = a\mathcal{N}(x; x', \ell^2 I)$ and the computation boils down to a convolution of two Gaussians:

$$\mathrm{e}^{t\mathcal{D}}k_{\mathrm{RBF}}(x, x') = \mathcal{N}(x; -t\beta, t\Sigma) * a\mathcal{N}(x; x', \ell^2 I) = a\mathcal{N}(x; x' - t\beta, \ell^2 I + t\Sigma) \tag{6}$$

$$\mathrm{e}^{t\mathcal{D}_1}k_{\mathrm{RBF}}(x, x')\mathrm{e}^{t\mathcal{D}_2'} = a\mathcal{N}(x; x' - t(\beta_1 - \beta_2), \ell^2 I + t\Sigma_1 + t\Sigma_2). \tag{7}$$

The application of the channel mixing matrices $W_k$ and summation is also straightforward through matrix multiplication for the mean and covariance. To summarize, because of the closed form action on the RBF kernel, the layer can be implemented efficiently and **exactly** with **no discretization** or approximations.

We note that with the Green's function above, the action of $\mathcal{A}$ encompasses the ordinary convolution operator on the 2d lattice as a special case. Given drift $\beta_k \in \{-1, 0, 1\}^{\times 2}, k = 1, \ldots, 9$ filling out the 9 elements of a $3 \times 3$ grid and as the diffusion $\Sigma_k \to 0$, the Green's function is a Dirac delta, so that: $\mathcal{A}f(x) = \sum_k W_k \delta(x - \beta_k) * f(x) = \sum_{i,j=-1,0,1} W_{ij} f(x_1 - i, x_2 - j) = W *_{\mathbb{Z}^2} f(x)$.

### 4.4 GENERAL EQUIVARIANCE

The convolutional layers discussed so far are translation equivariant. We discuss how to extend the continuous linear operator layers to more general symmetries such as rotations. Feature fields in this more general case are described by tensor fields, where the symmetry group acts not only on the input space $\mathcal{X}$ but also on the vector space attached to each point $x \in \mathcal{X}$. A linear layer $\mathcal{A}$ is equivariant if its action commutes with that of the symmetry. In appendix E we derive constraints for general linear operators and symmetries, which generalize those appearing in the steerable-CNN literature (Weiler and Cesa, 2019; Cohen et al., 2019). Then we show how equivariance under continuous roto-translations in 2d constrains the form of a convolutional layer by solving the equivariance constraint. Non-trivial solutions require that the operator $\mathcal{D}$ in the PDE of equation 3 has a non-trivial matrix structure.

## 4.5 Probabilistic Nonlinearities and Rectified Gaussian Processes

Gast and Roth (2018) derive the mean and variance for a univariate rectified Gaussian distribution for use in a neural network. We generalize these results to the full covariance function (and higher moments) of a rectified Gaussian process in appendix J and present the results here. For the input GP $\mathcal{A}^{(\ell)}f^{(\ell)}(x) \sim \mathcal{GP}(\mu(x), k(x, x'))$, we denote $\sigma(x) = \sqrt{k(x, x)}$, $\boldsymbol{\Sigma}$ the matrix with components $\Sigma_{ij} = k(x_i, x_j)$ for $i, j = 1, 2$ and $\boldsymbol{\mu} = [\mu(x_1), \mu(x_2)]$. We use notation $\Phi(z)$ for the univariate standard normal CDF, and $\boldsymbol{\Phi}(\boldsymbol{z}; \boldsymbol{\Sigma})$ for (two dimensional) multivariate CDF of $\mathcal{N}(0, \boldsymbol{\Sigma})$ at $\boldsymbol{z}$. $\boldsymbol{\Sigma}_1$ and $\boldsymbol{\Sigma}_2$ are the column vectors of $\boldsymbol{\Sigma}$. The first and second moments of $h = \text{ReLU}[\mathcal{A}f]$ are:

$$\mathbb{E}[h(x)] = \mu(x)\Phi(\mu(x)/\sigma(x)) + \sigma(x)\Phi'(\mu(x)/\sigma(x)), \tag{8}$$

$$\mathbb{E}[h(x_1)h(x_2)] = (k(x_1, x_2) + \mu(x_1)\mu(x_2))\boldsymbol{\Phi}(\boldsymbol{\mu}; \boldsymbol{\Sigma}) \tag{9}$$
$$+ (\mu(x_1)\boldsymbol{\Sigma}_2^\top + \mu(x_2)\boldsymbol{\Sigma}_1^\top)\nabla\boldsymbol{\Phi}(\boldsymbol{\mu}; \boldsymbol{\Sigma}) + \boldsymbol{\Sigma}_1^\top\nabla\nabla^\top\boldsymbol{\Phi}(\boldsymbol{\mu}; \boldsymbol{\Sigma})\boldsymbol{\Sigma}_2,$$

where $\nabla\nabla^\top\boldsymbol{\Phi}$ denotes the Hessian of $\boldsymbol{\Phi}$ with respect to the first argument. The first and higher order derivatives of the Normal CDF are just the PDF and products of the PDF with Hermite polynomials. Note that the mean and covariance interact through the nonlinearity.

## 4.6 Channel Mixing and Central Limit Theorem

After the non-linearity the process is no longer Gaussian. To overcome this issue we introduce a channel mixing matrix $M^{(\ell)} \in \mathbb{R}^{c_{\ell+1} \times c_\ell}$ and define the feature map in the following layer by $f^{(\ell+1)} = M^{(\ell)}h^{(\ell)}$, where $h^{(\ell)} = \text{ReLU}[\mathcal{A}^{(\ell)}f^{(\ell)}]$. So long as the channels of $h^{(\ell)}$ are only weakly dependent, we can apply the central limit theorem (CLT) to each function $f_\alpha^{(\ell+1)} = \sum_{\beta=1}^{c_\ell} M_{\alpha\beta}^{(\ell)}h_\beta^{(\ell)}$ so that in the limit of large $c_\ell$, the statistics of the $f_\alpha^{(\ell+1)}$'s converge to a GP with first and second moments given by:

$$\mathbb{E}[f^{(\ell+1)}(x)] = M\mathbb{E}[h^{(\ell)}(x)], \quad \mathbb{E}[f^{(\ell+1)}(x)f^{(\ell+1)}(x')^\top] = M\mathbb{E}[h^{(\ell)}(x)h^{(\ell)}(x')^\top]M^\top. \tag{10}$$

The convergence to a Gaussian process here is reminiscent of the well known infinite width limit of Bayesian neural networks Neal (1996); de G. Matthews et al. (2018); Yang (2019). However the setting here is fundamentally different. Unlike the Bayesian case where the distribution of $M$ is given by a prior or posterior, in PNCNN $M$ is a deterministic quantity and instead the uncertainty is about the input. PNCNN is not a Bayesian method in the sense of representing uncertainty about the parameters of the model, but instead it is Bayesian in representing and propagating the uncertainty in the value of the inputs. We go through the central limit theorem argument in appendix I, and go over some potential caveats such if the weights were to converge to $0$ or the independence assumptions are violated. In the following section, we evaluate the Gaussianity empirically.

## 4.7 Measurement and Projection to RBF Gaussian Process

As a last step we simplify the mean and covariance functions of the approximate GP $f^{(\ell+1)}$. While we can readily compute the values of these functions, unlike in the RBF kernel case, we cannot apply the convolution operator $e^{t\mathcal{D}}$ in closed form. In order to circumvent this challenge, we model the (approximately) Gaussian process $f^{(\ell+1)}$ with an RBF Gaussian process as follows: we evaluate the mean $y_i = \mathbb{E}[f^{(\ell+1)}(x_i)]$ and variance $\sigma_i^2 = \mathbb{V}\text{ar}[f^{(\ell+1)}(x_i)]$ of the approximate Gaussian process $f^{(\ell+1)}$ at a collection of points $\{x_i\}_{i=1}^N$ using equations 8, 9 and 10. These values $y_i$ are treated as measurements of the underlying signal with a heteroscedastic noise $\sigma_i^2$ that varies from point to point. We can then compute the RBF-based posterior GP of this signal $\hat{f}^{(\ell+1)}|\{(x_i, y_i, \sigma_i)\}_{i=1}^N \sim \mathcal{GP}(\mu_p, k_p)$ with posterior mean and covariance given by equation 1 for the heteroschedastic noise model. The uncertainty in the input $f^{(\ell)}$ is propagated through to the RBF posterior $\hat{f}^{(\ell+1)}|\{(x_i, y_i, \sigma_i)\}_{i=1}^N$ via the measurement noise $\sigma_i$. Crucially, this Gaussian process mean and covariance functions are written in terms of the RBF kernel and we can therefore continue applying convolutions in closed form in future layers.

As we describe in the following section, the RBF kernel in each layer is trained to maximize the marginal likelihood of the data that it sees, and thereby minimize the discrepancy with the underlying generating distribution $f^{(\ell+1)}$. While this measurement/projection approach is effective in

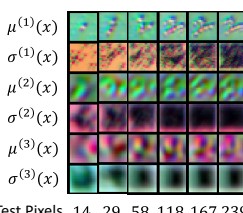 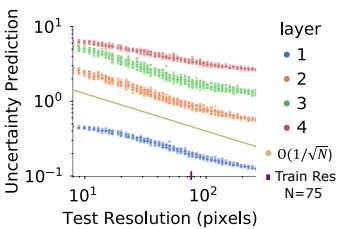 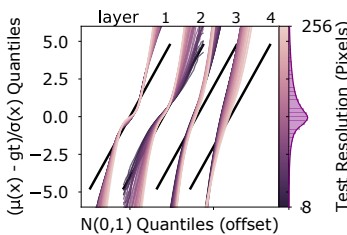

Figure 2: **Left:** Qualitative convergence of the mean and uncertainty of the first 3 channels of the feature maps is shown in RGB color as the input test resolution is increased. **Middle:** Median predicted uncertainties over spatial locations as a function of the test resolution. **Right:** Using the predictions of the highest resolution model as ground truth, the distribution of prediction residuals is shown in a Q-Q plot for each layer (shifted horizontally for clarity) with the black lines showing the theoretical relationship, and the overall distribution histogram is shown on the right.

many scenarios, in networks with many layers or a very large number of observations uncertainty information can get attenuated as it passes through the layers, a phenomenon which we investigate in appendix H.

With a network that is trained on a version of MNIST that is randomly subsampled to 75 pixels, in figure 2 (left) we evaluate the mean and uncertainty of the internal feature maps as we vary the number of pixels of the inputs at test time. As expected, the mean functions for the feature maps slowly converge and the predicted uncertainties decrease in magnitude as the input resolution is increased. In figure 2 (middle) we show that in early layers the uncertainties decrease at a similar rate to the $O(1/\sqrt{N})$ of discretization error that we would expect from a standard convolutional layer which is discretized to a square grid.[4] Despite the fact that these resolutions differ substantially from those seen at training time and the fact that there are no explicit uncertainty targets for these internal layers, the predictions are reasonably well calibrated as demonstrated in figure 2 (right). While the prediction residuals have fatter tails than a standard Gaussian (as evidenced by the deviations from the line in the QQ plot), the mean and standard deviation are close to the theoretically optimal 0 and 1 values across a range of resolutions.

### 4.8   Training procedure

Our neural network has two sets of parameters: the channel mixing and diffusion parameters, $\{(M^{(\ell)}, W^{(\ell)}, \beta^{(\ell)}, \Sigma^{(\ell)})\}_{\ell=1}^{L}$, as well as kernel hyperparameters of the Gaussian Processes $\{(l^{(\ell)}, a^{(\ell)})\}_{\ell=1}^{L}$. We train all parameters jointly on the loss $L_{\text{task}} + \lambda L_{\mathcal{GP}}$, where $L_{\text{task}}$ is the cross entropy with logits given by the mean $\mu_P$ of the pooled features $\mathcal{P}(f^{(L)}) \sim \mathcal{N}(\mu_P, \Sigma_P)$ and $L_{\mathcal{GP}}$ is the marginal log likelihoods of the GP feature maps:

$$L_{\mathcal{GP}}(f) = \frac{1}{2} \sum_{\ell=1}^{L} \sum_{\alpha=1}^{c_\ell} \left[ \left( \boldsymbol{f}_\alpha^T [K_{XX} + S_\alpha]^{-1} \boldsymbol{f}_\alpha \right) + \log \det \left[ K_{XX} + S_\alpha \right] + N \log 2\pi \right]^{(\ell)}, \quad (11)$$

where for each layer $\ell$, $\boldsymbol{f}_\alpha = [f_\alpha(x_1), ..., f_\alpha(x_N)] \in \mathbb{R}^N$ are the observed values for channel $\alpha$ at locations $X = [x_1, \ldots, x_N]$, $K_{XX}$ is the covariance of the RBF kernel and $S_\alpha = \text{diag}(\sigma_\alpha^2)$ the measurement noise for each channel $\alpha$ and spatial location. Notably the GP marginal likelihood is independent of the class labels.

## 5   Experimental Results

We evaluate the Probabilistic Numeric CNN on two different problems which have incomplete and irregular observations.

---

[4]A 2D discrete convolution layer using $N = m^2$ points can be interpreted as a Riemann sum approximation of the continuous integral and will therefore have an error of $O(1/m) = O(1/\sqrt{N})$ which is the same rate as would be achieved through Monte Carlo sampling.

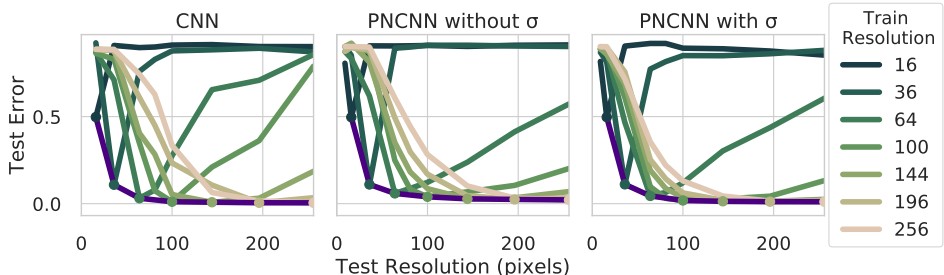

Figure 3: Zero shot generalization to other resolutions: Having trained on MNIST at a given training resolution shown by the color, we evaluate the performance of an PNCNN, PNCNN without uncertainty, and an ordinary CNN on varying test resolutions. Notably, PNCNN with uncertainty is the most robust.

**Superpixel MNIST** is an adaptation of the MNIST dataset where the 784 pixels of the original images are replaced by 75 salient *superpixels* that are non uniformly spread throughout the domain and are different for each image (Monti et al., 2017). Despite the simplicity of the underlying images, the lack of grid structure and high fraction of missing values make this a challenging task. Example inputs are visualized at the left of figure 1. We compare to Monet (Monti et al., 2017), SplineCNN (Fey et al., 2018), Graph Convolutional Gaussian Processes (GCGP) (Walker and Glocker, 2019), and Graph Attention Networks (GAT) (Avelar et al., 2020). As shown in table 5, the probabilistic numeric CNN greatly outperforms the competing methods reducing the classification error rate by more than $3\times$ over the previous state of the art.

| MNIST Superpixel 75 | Error Rate ($\downarrow$) | | PhysIONet2012 | AP ($\uparrow$) | AUROC ($\uparrow$) |
|---|---|---|---|---|---|
| Monet | 8.89 | | IP-Nets | 51.86 | 86.24 |
| SplineCNN | 4.78 | | SEFT-ATTN | 53.67 | 85.14 |
| GCGP | 4.2 | | GRU-D | **54.97** | **86.99** |
| GAT | 3.81 | | PNCNN | 53.92±.17 | 86.13±.06 |
| PNCNN | **1.24**±0.12 | | | | |
| PNCNN w/o $\sigma$ | 3.03±0.10 | | | | |

Table 1: Left: Classification Error on the 75-SuperPixel MNIST problem. Right: Average Precision and Area Under ROC curve metrics for PhysioNET2012. Mean and standard deviation are computed over 3 trials.

We conduct an ablation study where uncertainty propagation is removed: the probabilistic ReLU is replaced with the deterministic one applied to the mean, and the uncertainties in each layer are set to $0$. This form of the network still makes use of the fact that the input function is continuous by use of the GP interpolation of the means, but crucially it does not integrate the uncertainty in the computation resulting from the missing data. While this variant (PNCNN w/o $\sigma$) with a $3.03\%$ error rate outperforms existing methods from the literature, it is substantially worse than the PNCNN that integrates the uncertainty at $1.24\%$ error. This validates both that the underlying architecture (using the continuous convolution operators) has good inductive biases and that reasoning about discretization errors probabilistically can improve performance directly.

In figure 3 we evaluate the performance on zero shot generalization to a different test resolution for a variety of training resolutions. In order to compare to an ordinary CNN we sample MNIST on a regular square grid: PNCNN with uncertainty is the most robust to this train test sampling distribution shift, followed by PNCNN w/o uncertainty, and finally the ordinary CNN which is quite sensitive to these changes.

**Irregularly Spaced Time Series** For the second task, we evaluate our model on the irregularly spaced time series dataset PhysioNet2012 (Silva et al., 2012) for predicting mortality from ICU vitals signs. This dataset is particularly challenging because different vital sign channels are observed at different times, even within a single patient record. This means that we cannot compute the GP inference formula of equation 1 efficiently for all channels simultaneously because the observation

points $\{x_i\}$ and hence the matrices $K$ in that formula differ between the channels, increasing computational complexity. To circumvent this difficulty, we employ a stochastic diagonal estimator to compute the variances as described in appendix G. We compare against IP-Nets (Shukla and Marlin, 2019), SEFT-ATTN (Horn et al., 2019), and GRU-D (Che et al., 2018) as reported in Horn et al. (2019). PNCNN performs competitively, although not a breakout performance as in the image dataset which we attribute to the use of the stochastic variance estimates over an exact calculation.

## 6 CONCLUSION

Based on the ideas of probabilistic numerics, we have introduced a new class of neural networks which model missing values and discretization errors probabilistically *within the layers* of a CNN. The layers of our network are a series of operators defined on *continuous* functions, removing dependence on shortcut features like the sampling locations and distribution. On irregularly sampled and incomplete spatial data we show improved generalization and robustness. We believe that generative modeling of the internal feature maps of a neural network for representing uncertainty from incomplete observations is an underexplored research direction and has significant promise for other modalities of data, which we leave to future work.

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

## A  REVIEW OF GAUSSIAN PROCESSES

We briefly review here the main ideas of Gaussian Processes for machine learning, see Stein (2012); Rasmussen et al. (2006) for more details. We start to explain how to use stochastic processes for Bayesian inference. We see the stochastic process as prior over functions $p(f)$ and as we are given samples $\boldsymbol{x} = (x_1, \ldots, x_N), \boldsymbol{y} = (y_1, \ldots, y_N), y_i \equiv f(x_i)$, we update our beliefs about the function by constructing the posterior via Bayes rule: $p(f|\boldsymbol{y}, \boldsymbol{x}) = p(\boldsymbol{y}|f, \boldsymbol{x})p(f)/p(\boldsymbol{y}|\boldsymbol{x})$. Here we need to specify the likelihood of the data with our model $p(\boldsymbol{y}|f, \boldsymbol{x}) = \prod_{i=1}^{N} p(y_i|f, x_i)$ and the denominator, called the evidence or marginal likelihood, follows: $p(\boldsymbol{y}|\boldsymbol{x}) = \mathbb{E}_{f \sim p(f)}[p(\boldsymbol{y}|f, \boldsymbol{x})]$. The power of this approach is that the value of the signal $y$ at an unseen point $x$ has an uncertainty which depends on our knowledge of its neighbourhood: $p(y|x, \boldsymbol{y}, \boldsymbol{x}) = \mathbb{E}_{f \sim p(f|\boldsymbol{y}, \boldsymbol{x})}[p(y|f, x)]$ and allows us to reason probabilistically about the underlying signal.

A particular convenient class of random function is Gaussian processes (GPs) for which inference can be done exactly. A stochastic process can be presented in terms of the finite distributions of the random variables $\{f(x_i)\}_{i=1}^{M}$ at points $\{x_i\}_{i=1}^{M}$. For a GP these distributions are Gaussian and can be defined uniquely by specifying means and covariances, and so a GP is specified entirely by its mean function $\mu(x)$ and covariance kernel $k(x, x')$. We shall write $f \sim \mathcal{GP}(\mu, k)$. Let us assume a Gaussian likelihood model as well, i.e. $p(y_i|f, x_i) = \mathcal{N}(f(x_i), \sigma_i^2)$, where $\sigma_n$ represents aleatoric uncertainty on the measurement. (For simplicity we take here the function to be scalar valued but the reasoning can be easily generalized.) Then properties of the Gaussian distribution (see (Rasmussen et al., 2006, Chap. 2) for a detailed derivation of the formulas) lead to the following posterior distributions after seeing data $\boldsymbol{y}, \boldsymbol{x}$: $p(f|\boldsymbol{y}, \boldsymbol{x}) = \mathcal{GP}(\mu_p, k_p)$, with

$$\mu_p(x) = \boldsymbol{k}(x)^T[K + S]^{-1}\boldsymbol{y}, \quad k_p(x, x') = k(x, x') - \boldsymbol{k}(x)^T[K + S]^{-1}\boldsymbol{k}(x'). \tag{12}$$

where $K_{ij} = k(x_i, x_j), k(x)_i = k(x, x_i)$ and $S = \mathrm{diag}(\sigma_i^2)$.

## B  FROM DISCRETE TO CONTINUOUS CONVOLUTIONAL LAYERS

We here show that the general formula

$$\mathcal{A} = \sum_k W_k \mathrm{e}^{\mathcal{D}_k} \tag{13}$$

with $\mathcal{D}_k$ a function of spatial derivatives, reduces in the case of discrete input space $\mathcal{X}$ to the usual convolution we encounter in deep learning.

For simplicity we shall assume a 1d grid as input space $\mathcal{X} = \{1, \ldots, N\}$. Let us start by recalling the form of the classical discrete convolution when $C_\ell = C_{\ell+1} = 1$. We define a convolutional layer as a linear map that commutes with the translation operator. To make the symmetry exact, we need assume periodic boundaries. Then in the standard basis of $\mathbb{R}^N$, $\{e_i\}_{i=1}^N$ of vectors localized at site $i$, the translation operator $\tau$ acts as $\tau e_i = e_{i+1 \mod N}$. An $N \times N$ matrix $B$ is translation invariant iff $\tau B = B\tau$. Since $\tau$ is diagonal in Fourier space, the most general solution is $B = F^{-1}\mathrm{diag}(\hat{\boldsymbol{b}})F$, where $F_{jk} = \mathrm{e}^{-\frac{2\pi i}{N}jk}$ is the discrete Fourier transform. Such matrices are called circulant and can be written alternatively as $B = \sum_{i=0}^{N-1} b_{N-i}\tau^i, \boldsymbol{b} = F^{-1}\hat{\boldsymbol{b}}$. Explicitly:

$$B = \begin{pmatrix} b_0 & b_1 & \ldots & b_{N-2} & b_{N-1} \\ b_{N-1} & b_0 & b_1 & & b_{N-2} \\ \vdots & b_{N-1} & b_0 & \ddots & \vdots \\ b_2 & & \ddots & \ddots & b_1 \\ b_1 & b_2 & \ldots & b_{N-1} & b_0 \end{pmatrix}. \tag{14}$$

This shows that the most general convolutional layer is a circulant matrix. E.g. if $b_i = 0$ unless $i = 0, 1, N-1$, $B$ coincides with the matrix representing a periodic convolution of filter size 3. The matrix $B$ is invertible as long as $\hat{b}_k \neq 0$ for all $k$. In a convolutional network the parameters $b_i$ are random variables and the measure of the set where $B$ is not invertible is zero. Thus the role of $\mathrm{e}^{\mathcal{D}}$ is replaced in the discrete case by the $B$.

The discrete analog of $\mathcal{A}$ is then:

$$A = \sum_i W_i \otimes B_i \,. \tag{15}$$

Introducing the unit matrices $E_{\alpha,\beta}$ which have 1 at the row $\alpha$ and column $\beta$ and 0 otherwise, we can rewrite it as:

$$A = \sum_{j,\alpha,\beta} E_{\alpha,\beta} \otimes \tau^j W_j^{\alpha,\beta}, \quad W_j^{\alpha,\beta} = \sum_i W_i^{\alpha,\beta} b_{i,N-j} \,. \tag{16}$$

Since $E_{\alpha,\beta} \otimes \tau^j$ is a linear basis of the space of convolutional layers, we see that equation 13 indeed reduces to the usual one when discretizing the input domain and is a principled generalization to the continuous domain.

## C  GREENS FUNCTION

Given the operator $\mathcal{D} = \beta^\top \nabla + \frac{1}{2} \nabla^\top \Sigma \nabla$ we can compute the action of $e^{t\mathcal{D}}$ in terms of convolutions. Using the $d$ dimensional Fourier transforms $\mathcal{F}[h](k) = (2\pi)^{-\frac{d}{2}} \int h(x) e^{-ik^\top x} dx$ and $\mathcal{F}^{-1} = \mathcal{F}^\dagger$, we can rewrite the derivative operator $\mathcal{D}$ in terms of elementwise multiplication in the Fourier domain, which diagonalizes $\mathcal{D}$. Since $\nabla = \mathcal{F}^{-1}(ik)\mathcal{F}$,

$$\mathcal{D} = \mathcal{F}^{-1}(i\beta^\top k - \tfrac{1}{2} k^\top \Sigma k)\mathcal{F}. \tag{17}$$

Using the series definition $e^{t\mathcal{D}} = \sum_{n=0}^\infty (t\mathcal{D})^n / n!$, we have:

$$e^{t\mathcal{D}} = \mathcal{F}^{-1} e^{t(i\beta^\top k - \frac{1}{2} k^\top \Sigma k)} \mathcal{F}. \tag{18}$$

Applying this operator to a test function $h(x)$ yields

$$e^{t\mathcal{D}} h = \mathcal{F}^{-1}[e^{t(i\beta^\top k - \frac{1}{2} k^\top \Sigma k)} \mathcal{F}[h](k)] = \mathcal{F}^{-1}[\mathcal{F}[G_t] \cdot \mathcal{F}[h]] = G_t * h, \tag{19}$$

where the final step follows from the Fourier convolution theorem, and we define the function $G_t = \mathcal{F}^{-1}[e^{t(i\beta^\top k - \frac{1}{2} k^\top \Sigma k)}]$. Directly applying the Fourier integral yields a Gaussian integral

$$G_t(x) = (2\pi)^{-\frac{d}{2}} \int e^{ik^\top (x+t\beta) - \frac{1}{2} k^\top t\Sigma k} dk = e^{-\frac{1}{2}(x+t\beta)^\top (t\Sigma)^{-1}(x+t\beta)} \det(2\pi t\Sigma)^{-1/2}. \tag{20}$$

This function $G_t(x) = \mathcal{N}(x; -t\beta, t\Sigma)$ is nothing but a multivariate heat kernel, the Greens function (also known as the fundamental solution or time propagator) for the diffusion equation $\partial_t G_t(x - x') = \mathcal{D} G_t(x - x')$, and indeed $\lim_{t\to 0} G_t(x - x') = \delta(x - x')$.

## D  INTEGRAL POOLING

The integral pooling operator $\mathcal{P}[f] = \int_{\mathbb{R}^d} f(x) dx$ can be applied to the Gaussian process just like any other linear operator. Given $f^{(L)} \sim \mathcal{GP}(\mu, k)$, we have that

$$\mathcal{P} f^{(L)} \sim \mathcal{GP}(\mathcal{P}\mu, \mathcal{P}k\mathcal{P}') = \mathcal{N}(\mathcal{P}\mu, \mathcal{P}k\mathcal{P}'). \tag{21}$$

Again, computing the mean $\mu_P = \mathcal{P}\mu$ and covariance matrix $\Sigma_P = \mathcal{P}k\mathcal{P}'$ we need just to be able to apply $\mathcal{P}$ to the RBF kernel.

$$\mathcal{P}k_{\mathrm{RBF}}(x') = \int_{\mathbb{R}^d} k_{\mathrm{RBF}}(x, x') dx = a \tag{22}$$

$$\mathcal{P}k_{\mathrm{RBF}}\mathcal{P}' = \int_{\mathbb{R}^d \times \mathbb{R}^d} k_{\mathrm{RBF}}(x, x') dx dx' = \infty \tag{23}$$

For many applications such as image classification using the mean logit value, we require only the predictive mean, so an unbounded covariance matrix $\Sigma_P$ is acceptable. We use this form for all of our experiments.

However for some applications an output uncertainty can be useful, so we also provide a variant that integrates over a finite region $[0,1]^d$, $\mathcal{P}f = \int_{[0,1]^d} f(x)dx$.

$$\mathcal{P}k_{\mathrm{RBF}}(x') = \int_{[0,1]^d} k_{\mathrm{RBF}}(x,x')dx = a\prod_{i=1}^{d}\left[\Phi(\tfrac{x'_i}{\ell}) - \Phi(\tfrac{x'_i-1}{\ell})\right] \tag{24}$$

$$\mathcal{P}k_{\mathrm{RBF}}\mathcal{P}' = \int_{[0,1]^d\times[0,1]^d} k_{\mathrm{RBF}}(x,x')dxdx' = a\left[\ell\sqrt{\tfrac{2}{\pi}}(e^{-1/2\ell^2}-1) + 2\Phi(\tfrac{1}{\ell}) - 1\right]^d \tag{25}$$

where $\Phi$ is again the univariate standard normal CDF.

## E    EQUIVARIANCE

### E.1    RELATED WORK

We note that there has been considerable research effort in the development of equivariant CNNs which we build on top of. The group equivariant CNN was introduced by Cohen and Welling (2016a) for discrete groups on lattices. This work has been extended for continuous groups (Worrall et al., 2017; Zhou et al., 2017) and with steerable equivariance (Cohen and Welling, 2016b; Weiler and Cesa, 2019) where other group representations are used. There have also been group equivariant networks designed for point clouds and other irregularly spaced data (Thomas et al., 2018; Finzi et al., 2020; Fuchs et al., 2020; de Haan et al., 2020). In Shen et al. (2020), layers using finite difference estimation of derivative operators are used for defining equivariant layers in an equivariant CNN, effectively a change of basis.

Most closely related to our PDE operator approach to equivariance is work by Smets et al. (2020). In this work, the authors define layers of their convolutional network through the time evolution of a PDE which is a nonlinear generalization of the diffusion equation, and which additionally includes pooling like behaviour. Smets et al. (2020) parametrize their equivariant PDEs in terms of left invariant vector fields. These PDEs are applied to scalar functions on the group, unlike our work where feature fields (of various representations) live on the base space $\mathbb{R}^n$. These two approaches reflect the two major approaches to equivariance in the literature: using regular representations and using irreducible/tensor representations. Both have advantages and disadvantages and we focus on the second approach in this work. It may be possible that there is a deeper relationship between the solutions we find and the left invariant vector fields in Smets et al. (2020), but this is beyond the scope of our work.

### E.2    TRANSLATION EQUIVARIANCE

A key factor in the generalization of convolutional neural networks is their translation equivariance. Patterns in different parts of an input signal can be seen in the same way because convolution is translation equivariant. Our learnable linear operators $\mathcal{A}$ are equivariant to continuous transformations. Two linear operators $e^{\mathcal{C}}$ and $\mathrm{e}^{\mathcal{B}}$ commute $[\mathrm{e}^{\mathcal{C}}, \mathrm{e}^{\mathcal{B}}] = 0$ if and only if their generators commute: $[\mathcal{C}, \mathcal{B}] = 0$. Since the generator of diffusions $\mathcal{D}_i$ is a sum of derivative operators, and the generators of translations are just $\nabla$ as mentioned in section 4.2, the two commute: $[\mathcal{D}_k, \nabla] = 0$. Therefore $[\mathcal{A}, \tau_a] = [\sum_k W_k \mathrm{e}^{\mathcal{D}_k}, \tau_a] = \sum_k W_k[\mathrm{e}^{\mathcal{D}_k}, \mathrm{e}^{a^\top \nabla}] = 0$ and $\mathcal{A}$ is translation equivariant.

### E.3    STEERABLE EQUIVARIANCE FOR LINEAR OPERATORS

For some tasks like medical segmentation, aerial imaging, and chemical property prediction there are additional symmetries in the data it makes sense to exploit other than mere translation equivariance. Below we show how to enforce equivariance of the Linear operator $\mathcal{A}$ to other symmetry groups $G$ such as the group of continuous rotations $\mathrm{SO}(d)$ in $\mathbb{R}^d$. Applying equivariance constraints separately on each of the components of $\mathrm{e}^{\mathcal{D}_i}$ on top of translation equivariance yields very restricted set of operators. For example, enforcing equivariance to continuous rotations $G = \mathrm{SO}(d)$, the operator must be an isotropic heat kernel: $\mathcal{D}_k = c_k \nabla^\top \nabla$. The reason for this apparent restriction is a result of considering the different channels independently, as scalar fields.

The alternative is to use features fields which transform under more general representations of the symmetry group, introduced in steerable-CNNs (Cohen and Welling, 2016b) and used in (Worrall

et al., 2017; Thomas et al., 2018; Weiler et al., 2018; Weiler and Cesa, 2019) and others. In this way, the symmetry transformation acts not only on the spatial domain $\mathcal{X}$, but also transforms the channels. The way that the group acts on $\mathbb{R}^c$ (i.e. the channels) is formalized by a representation matrix $\rho(g) \in \mathbb{R}^{c \times c}$ for each element $g \in G$ in the transformation group that satisfies $\forall g, h \in G :$ $\rho(gh) = \rho(g)\rho(h)$. Choosing the type of each intermediate feature map is equivalent to choosing their representations, and we describe a simple way of doing this with tensor representations in the later section.

**Operator Equivariance Constraint**: Returning to linear operators, we derive the equivariance constraint and show how to use constructs from the previous sections to implement steerable rotation equivariance. Equivariance of a linear operator $\mathcal{A} : \left(\mathbb{R}^d \to \mathbb{R}^{c_{in}}\right) \to \left(\mathbb{R}^d \to \mathbb{R}^{c_{out}}\right)$ requires that, for any input function, transforming the input function first (both argument and channels) and applying $\mathcal{A}$ is equivalent to first applying $\mathcal{A}$ and then transforming the output: $\mathcal{A}\rho_{in}(g)L_g f = \rho_{out}(g)L_g \mathcal{A}f$ where $L_g f(x) = f(g^{-1}x)$. Rearranging the terms, one sees that the equivariance constraint on the linear operator $\mathcal{A}$ is:

$$\rho_{out}(g)L_g \mathcal{A} L_{g^{-1}} \rho_{in}(g^{-1}) = \mathcal{A}, \tag{26}$$

where the operators $L_g$ and $L_g^{-1}$ are understood not to act on the representation matrices $\rho$ (although implicitly a function of $g$). As shown in Appendix E.4, eq. 26 is a direct generalization of the equivariance constraint for convolutions $\forall x : \rho_{out}(g)K(g^{-1}x)\rho_{in}(g^{-1}) = K(x)$ described in the literature (Weiler and Cesa, 2019; Cohen et al., 2019).

As shown in Appendix E.6, the equivariance constraint for continuous rotations applied to the diffusion operators $\mathcal{A} = \sum_k W_k e^{\mathcal{D}_k}$ has only the trivial solutions of isotropic diffusion without any drift. For this reason we instead consider a more general form of diffusion operator where the PDE itself couples the different channels. For the coupled PDE:

$$\frac{\partial f}{\partial t} = \sum_k W_k \mathcal{D}_k f \tag{27}$$

the time evolution contains the matrices $W_k$ *in* the exponential $\mathcal{A} = e^{\sum_k W_k \mathcal{D}_k}$. Like with the example of translation above, this operator is equivariant if and only if the infinitesmal generator $\sum_k W_k \mathcal{D}_k$ is equivariant.

Because equation 26 applies generally to linear operators and not just convolutions, we can compute the equivariance constraint for these derivative operators. We can simplify the summation $\sum_k W_k \mathcal{D}_k = \sum_k W_k(\beta_k^T \nabla + (1/2)\nabla^T \Sigma_k \nabla)$ by writing it in terms of the collection of matrices $B_i = \sum_k W_k \beta_{ki}$ and $S_{ij} = (1/2)\sum_k W_k \Sigma_{kij}$ to express $\mathcal{A}_{\text{deriv}} = \sum_i B_i \partial_i + \sum_{i,j} S_{ij}\partial_i\partial_j$ where the indices $i, j = 1, 2..., d$ enumerate the spatial dimensions of each vector $\beta_k$ and each matrix $\Sigma_k$. As we derive in appendix E.5, the necessary and sufficient conditions for the equivariance of $\sum_k W_k \mathcal{D}_k$ and therefore $\mathcal{A}$ is that $\forall g \in G : [\rho_{out} \otimes \rho_{in}^* \otimes \rho_{(1,0)}](g)\text{vec}(B) = \text{vec}(B)$ and $\forall g \in G : [\rho_{out} \otimes \rho_{in}^* \otimes \rho_{(2,0)}](g)\text{vec}(S) = \text{vec}(S)$ where $\text{vec}(\cdot)$ denotes flattening the elements into a single vector and $\rho_{(r,s)}$ is the tensor representation with $r$ covariant and $s$ contravariant indices.

### E.4 GENERALIZATION OF EQUIVARIANCE CONSTRAINT FOR CONVOLUTIONS

This equivariance constraint is a direct generalization of the equivariance constraint for convolution kernels as described in Weiler and Cesa (2019); Cohen et al. (2019). In fact, when $\mathcal{A}$ is a *convolution* operator, $\mathcal{A}f = K * f$, the action of $L_g$ by conjugation $\mathcal{A}$ is equivalent to transforming the argument of the kernel $K$:

$$L_g(K*)L_{g^{-1}}f(x) = \int K(g^{-1}x - x')f(gx')d\mu(x')$$

$$= \int K(g^{-1}(x - x''))f(x'')d\mu(x'') = (L_g[K]) * f.$$

Letting both sides of eq 26 act on the product of a constant unit vector $e_i$ and a delta function, $f = e_i\delta$ the expression $\forall e_i : \rho_{out}(g)L_g[K]\rho_{in}(g^{-1}) * e_i\delta = K * e_i\delta$ can be rewritten as $\forall x : \rho_{out}(g)K(g^{-1}x)\rho_{in}(g^{-1}) = K(x)$ which is precisely the constraint for steerable equivariance for convolution described in the literature. [5]

---

[5] This assumes as is typically done that measure $\mu$ over which the convolution is performed is left invariant. For the more general case, see the discussion in Bekkers (2019).

## E.5 EQUIVARIANT DIFFUSIONS WITH MATRIX EXPONENTIAL

Below we solve for the **necessary** and **sufficient** conditions for the equivariance of the operator $\mathcal{A}_{\text{deriv}}$.

We will use tensor representations for their convenience, but the approach is general to allow other kinds of representations. A rank $(p, q)$ tensor $t$ is an element of the vector space $T_{(p,q)} := V^{\otimes p} \otimes (V^*)^{\otimes q}$ where $V$ is some underlying vector space, $V^*$ is its dual and $(\cdot)^{\otimes p}$ is the tensor product iterated $p$ times. In common language $T_{(0,0)}$ are scalars, $T_{(1,0)}$ are vectors, and $T_{(1,1)}$ are matrices. Given the action of a group $G$ on the vector space $V$, the representation on $T_{(p,q)}$ is $\rho_{(p,q)}(g) = g^{\otimes p} \otimes (g^{-\top})^{\otimes q}$ where $-\top$ is inverse transpose and $\otimes$ on the matrices is the tensor product (Kronecker product) of matrices. Composite representations can be formed by stacking different tensor ranks together, such as a representation of 50 scalars, 25 vectors, 10 matrices and 5 higher order tensors: $T_{(0,0)}^{50} \oplus T_{(1,0)}^{25} \oplus T_{(1,1)}^{10} \oplus T_{(1,2)}^{5}$, where $\oplus$ in this context is the same as the Cartesian product. For a composite representation $U = \bigoplus_i T_{(p_i, q_i)}$ the group representation is similarly $\rho_U(g) = \bigoplus_i \rho_{(p_i, q_i)}(g)$ where $\oplus$ concatenates matrices as blocks on the diagonal.

Noting that the operator $L_g$ that acts only on the argument and the matrix $\rho_{in}(g)$ acts only on the components, the two commute and we can rewrite the constraint for $\mathcal{A}_{\text{deriv}}$ as

$$\sum_i \rho_{out}(g) B_i \rho_{in}(g^{-1}) L_g \partial_i L_{g^{-1}} + \sum_{ij} \rho_{out}(g) S_{ij} \rho_{in}(g^{-1}) L_g \partial_i \partial_j L_{g^{-1}} = \mathcal{A}_{\text{deriv}} \quad (28)$$

We can simplify the expression $L_g \partial_i L_{g^{-1}}$ by seeing how it acts on a function. For any differentiable function $\partial_i L_{g^{-1}} f(x) = \frac{\partial}{\partial x_i} [f(gx)] = \sum_j g_{ji} [\partial_j f](gx) = L_{g^{-1}} \sum_j g_{ji} \partial_j f(x)$ where $g_{ij}$ are the components of the matrix $g$. Since this holds for any $f$, we find that $L_g \nabla L_{g^{-1}} = g^T \nabla$ and therefore $L_g \nabla \nabla^T L_{g^{-1}} = L_g \nabla L_{g^{-1}} L_g \nabla^T L_{g^{-1}} = g^T \nabla \nabla^T g$.

Since equation 28 holds as an operator equation, it must be true separately for each component $\partial_i$ and $\partial_i \partial_j$. This means that the constraint separates into a constraint for $B$ and a constraint for $S$:

1. $\forall g, i : \sum_j g_{ij} \rho_{out}(g) B_j \rho_{in}(g^{-1}) = B_i$
2. $\forall g, i, j : \sum_{kl} g_{i\ell} g_{ik} \rho_{out}(g) S_{\ell k} \rho_{in}(g^{-1}) = S_{ij}$.

These relationships can be expressed more succinctly by flattening the elements of $B$ and $S$ into vectors: $[\rho_{out}(g) \otimes \rho_{in}(g^{-T}) \otimes \rho_{(1,0)}(g)] \text{vec}(B) = \text{vec}(B)$ and $[\rho_{out}(g) \otimes \rho_{in}(g^{-T}) \otimes \rho_{(2,0)}(g)] \text{vec}(S) = \text{vec}(S)$.

## E.6 ROTATION EQUIVARIANCE CONSTRAINT FOR SCALAR DIFFUSIONS HAS ONLY TRIVIAL SOLUTIONS

The diffusion operator $\mathcal{A} = \sum_k W_k e^{\mathcal{D}_k}$ leads to only trivial $\beta_k = 0$ and $\Sigma_k \propto I$ if it satisfies the continuous rotation equivariance constraint.

**Proof:**

The application of $e^{\mathcal{D}_k}$ is just a convolution with the Greens function

$$\sum_k W_k e^{\mathcal{D}_k} f = \sum_k W_k [e^{-\frac{1}{2}(x+\beta_k)^\top \Sigma_k^{-1}(x+\beta_k)} \det(2\pi \Sigma_k)^{-1/2}] * f = \sum_k W_k G_k * f \quad (29)$$

where the Greens function is the multivariate Gaussian density: $G_k(x) = \mathcal{N}(x; -\beta_k, \Sigma_k)$.

As shown in appendix E.4, for convolutions the operator constraint is equivalent to the kernel equivariance constraint $\rho_{out}(g) K(g^{-1} x) \rho_{in}(g^{-1}) = K(x)$ from (Weiler and Cesa, 2019). With $K(x) = \sum_k W_k G_k(x)$ this reads:

$$\forall x \in \mathbb{R}^d, g \in G: \quad \sum_k \rho_{out}(g) W_k \mathcal{N}(g^{-1} x; -\beta_k, \Sigma_k) \rho_{in}(g^{-1}) = \sum_k W_k \mathcal{N}(x; -\beta_k, \Sigma_k),$$

For rotations $g \in SO(2)$ where we can parametrize $g_\theta = e^{\theta J}$ in terms of the antisymmetric matrix $J = [[0, 1], [-1, 0]] \in \mathbb{R}^{2 \times 2}$ and the translation operator can be written $L_g = e^{-\theta x^T J^T \nabla}$, we can

take derivatives with respect to $\theta$ to get (now with double sums implicit):

$$\forall x \in \mathbb{R}^d : \sum_k \big[ d\rho_{out} W_k \mathcal{N}(x; -\beta_k, \Sigma_k) - W_k \mathcal{N}(x; -\beta_k, \Sigma_k) d\rho_{in} - W_k(x^T J^T \nabla) \mathcal{N}(x; -\beta_k, \Sigma_k) \big] = 0.$$

Here the Lie Algebra representation of $J$ is $d\rho := \frac{\partial}{\partial\theta} \rho(g_\theta)|_{\theta=0}$. Factoring out the normal density:

$$\forall x \in \mathbb{R}^d : \quad \sum_k \big[ d\rho_{out} W_k - W_k d\rho_{in} - W_k(x^T J^T \Sigma_k^{-1}(x + \beta_k)) \big] \mathcal{N}(x; -\beta_k, \Sigma_k) = 0.$$

Without loss of generality we may assume that each of the Gaussians $\beta_k, \Sigma_k$ pairs are distinct since if they were not then we could replace the collection with a single element. Since the (finite) sum of distinct Gaussian densities is never a Gaussian density, and monomials of order $> 0$ multiplied by a Gaussian density cannot be formed with sums of Gaussian densities or sums multiplied by monomials of a different order and Gaussian densities are never 0, this constraint separates out into several independent constraints.

1. $\forall i : d\rho_{out} W_k = W_k d\rho_{in}$
2. $\forall i, x : W_k(x^T J^T \Sigma_k^{-1} \beta_k) = 0$
3. $\forall i, x : W_k(x^T J^T \Sigma_k^{-1} x) = 0$

We may assume w.l.o.g. that $W_k$ is not 0 for all components of the matrix (otherwise we could have deleted this element of $k$ and continue). Therefore there is some component which is nonzero, and the expressions in parentheses in equations 2 and 3 must be 0. Given that this holds for all $x$, eq 3 implies: $J^T \Sigma_k^{-1} = 0$ or equivalently $\Sigma_k^{-1} J = 0$ because $\Sigma_k$ is symmetric, and since $J = -J^T$ this can be expressed concisely as $[\Sigma_k^{-1}, J] = 0$ for which the only symmetric solution is proportional to the identity $\Sigma_k = c_k I$. Since both $\Sigma_k$ and $J$ are invertible, equation 2 yields $\beta_k = 0$. Therefore there are no nontrivial solutions for $\beta, \Sigma$ in $\mathcal{A} = \sum_k W_k e^{\mathcal{D}_k}$ for continuous rotation equivariance.

## F  DATASET AND TRAINING DETAILS

In this section we elaborate on some of the details regarding hyperparameters, network architecture, and the datasets.

As described in the main text, the PNCNN is composed of a chain of convolutional blocks containing a convolution layer, a probabilistic ReLUs, and linear channel mixing layer (analogue of the colloquial $1 \times 1$ convolution). In each of these convolutional blocks, the input is a collection of points and feature mean value at those points along with the feature elementwise standard deviation at those points: $\{(x_i, \mu(x_i), \sigma(x_i)\}_{i=1}^N$. These observations seed the GP layer, and the block is evaluated at the same collection of points for the output (although it can be evaluated elsewhere since it is a continuous process, and we make use this fact to visualize the features in figures 1 and 2).

**Hyperparameters:** For the PNCNN on the Superpixel MNIST dataset, we use 4 PNCNN convolution blocks with $c = 128$ channels and with $K = 9$ basis elements for the different drift and diffusion parameters in $\sum_{k=1}^K W_k e^{\mathcal{D}_k}$. We train for 20 epochs using the Adam optimizer (Kingma and Ba, 2014) with $\mathrm{lr} = 310^{-3}$ with batch size 50.

For the PNCNN on the PhysioNet2012 dataset, we use the variant of the PNCNN convolution layer that uses the stochastic diagonal estimator described in appendix G with $P = 20$ probes. In the convolution blocks we use $c = 96$ channels, $K = 5$ basis elements and we train for 10 epochs using the same optimizer settings above. For both datasets we tuned hyperparameters on a validation set of size $10\%$ before folding the validation set back into the training set for the final runs. Both models take about 2 hours to train.

**SuperPixel-MNIST** We source the SuperPixel MNIST dataset (Monti et al., 2017) from Fey and Lenssen (2019) consisting of $60k$ training examples and $10k$ test represented as collections of positions and grayscale values $\{(x_i, f(x_i))\}_{i=1}^{75}$ at the $N = 75$ super pixel centroids.

**PhysioNet2012** We follow the data preprocessing from Horn et al. (2019) and the $10k$-$2k$ train test split. The individual data points consist of 42 irregularly spaced vital sign time series signals as

well as 5 static variables: Gender, ICU Type, Age, Height, Weight. We use one hot embeddings for the first two categoric variables, and we treat each of these static signals as fully observed constant time series signals. As the binary classification task exhibits a strong label imbalance, $14\%$ positive signals, we apply an inverse frequency weighting of $1/.14$ to the binary cross entropy loss.

## G   STOCHASTIC DIAGONAL ESTIMATION FOR PHYSIONET2012

In order to compute the mean and variance of the rectified Gaussian process, the activations of the probabilistic ReLU, we need compute the diagonal of $\mathcal{A}k_p\mathcal{A}'(x_n, x_n)$ for the relevant points $\{x_n\}_{n=1}^N$.

In the usual case where each of the channels $\alpha = 1, 2, ..., c$ are observed at the same locations this can be done efficiently. First one computes the application of $e^{\mathcal{D}_i}$ on the left and $e^{\mathcal{D}'_j}$ on the right onto the posterior $k_p$:

$$N_{ij} = (e^{\mathcal{D}_i} k_p e^{\mathcal{D}'_j})(x_n, x_n) = (e^{\mathcal{D}_i} k e^{\mathcal{D}'_j})(x_n, x_n) - (e^{\mathcal{D}_i}\mathbf{k}^\top)(x_n)[K + S]^{-1}(\mathbf{k}e^{\mathcal{D}'_j})(x_n)$$

where $k$ is the RBF kernel and we have reused the notation from appendix A. Notably, this quantity is the same for each of the channels, and the elementwise variance is just:

$$v_\alpha(x_n) = (\mathcal{A}k_p\mathcal{A}')_{\alpha\alpha}(x_n, x_n) = \sum_{i,j,\beta} W_i^{\alpha\beta} N_{ij} W_j^{\alpha\beta} \tag{30}$$

where the $\alpha, \beta$ index the channels of each of the matrices $W_i$. Because $N$ is the same for all channels, we can compute this quantity efficiently with a reasonable memory cost and compute.

For the PhysioNet2012 dataset where the observation points differ between the channels we must consider a different observation set $\{x_n^\beta\}_{n=1}^N$ for each channel $\beta$. This means that evaluated kernel depends on the channel and we have the objects: $\mathbf{k}^\beta$, $K^\beta$ and $S^\beta$. As a result, we have an additional index for $N_{ij}^\beta$ and the desired computation is

$$v_\alpha(x_n^\alpha) = (\mathcal{A}k_p\mathcal{A}')_{\alpha\alpha}(x_n^\alpha, x_n^\alpha) = \sum_{i,j,\beta} W_i^{\alpha\beta} N_{ij}^\beta W_j^{\alpha\beta}. \tag{31}$$

While each of the terms in the computation can be computed without much difficulty, performing the summation explicitly requires an unreasonably large memory cost and also compute.

However, by the same approach we can consider the full covariance matrix $\mathbf{B}_{(\alpha n)(\beta m)} = (\mathcal{A}k_p\mathcal{A}')_{\alpha\beta}(x_n^\alpha, x_m^\beta)$, and while it would not be feasible to compute this matrix directly we *can* define matrix vector multiplies onto vectors of size $\mathbb{R}^{cN}$ implicitly using the sequence of operations that define it. Crucially, this sequence of operations has much more modest memory consumption (and compute cost) over the direct expression in equation 31. These implicit matrix vector multiplies can then be used to compute a stochastic diagonal estimator (Bekas et al., 2007) given by:

$$\hat{v}_\alpha(x_n^\alpha) = \frac{1}{P}\sum_{p=1}^P z_p \odot \mathbf{B}z_p \tag{32}$$

with Gaussian probe vectors $z_p \sim \mathcal{N}(0, I)$, and where $\odot$ is elementwise multiplication (see Bekas et al. (2007) for more details on this stochastic diagonal estimator). We use this estimator with $P = 20$ probes for computing the variances for PhysioNet. We note that with $P = 20$ the variance estimates are still quite noisy, however without the estimator cannot readily apply the PNCNN to PhysioNet. We leave a better approach for handling this kind of data to future work.

## H   PATHOLOGIES IN PROJECTION TO RBF GAUSSIAN PROCESS

In section 4.7 describe an approach by which a Gaussian process with a complex mean and co-variance function is *projected* down to the posterior of a (simpler) RBF kernel GP from a set of observations. We know given the representation capacity of the RBF kernel that with the right set of observations, a complex function can be well approximated in principle. However, the relationship for uncertainty is less straightforward.

The properties of the input Gaussian process must be conveyed to the output Gaussian process by only the (uncorrelated) noisy observations $\{(x_i, \mu(x_i), \sigma(x_i))\}_{i=1}^N$. As the uncertainty in original GP increases, so do the measurement uncertainties in the transmission, and therefore the output GP also has a higher uncertainty. However, the uncertainty in the input GP is in the form of a full covariance kernel $k(x, x')$ and it seems that individual observations will not easily be able to communicate the *covariance* of the values of the GP function at different spatial locations despite the heterogeneous noise model.

Fundamentally, the problem is that the observation values are treated as independent, an incorrect assumption which has other knock-on effects when the number of observations is large. With some fixed measurement error no matter how high but a large enough set of independent observations, the mean value can be pinned down precisely. If in contrast the observations are not independent, then there may be a situation where the mean value cannot be known more precisely than some limiting uncertainty. This effect leads the output GP to have less uncertainty and be more confident in the values that it should be given the input GP.

If the observations are sparse, then the effective sample size of the estimator for the mean of the GP at any given location is small, and then the amount by which uncertainty is underestimated is small. However, if there are very many observations then this kind of observation transmission of information with the independence assumption will attenuate the uncertainty. We would also expect that over the course of many layers, this attenuation can accumulate. We believe that this is what causes the poorer uncertainty calibration in layers 3 and 4 of the PNCNN shown in figure 2. We hope that this problem can be resolved perhaps by removing the independence assumption or providing an alternative projection method in future work.

# I  CENTRAL LIMIT THEOREM FOR STOCHASTIC PROCESSES

In this section we show that each of the outputs features $f_\alpha = \sum_\beta M_{\alpha\beta} h_\beta$ are marginally distributed as Gaussian processes in the limit as $c_{in} \to \infty$, provided that $h_\beta$ are independent and have bounded moments, and that the fraction of zeros in each row of $M$ does approach 1. The result follows from an application of the multivariate Lyapunov CLT to the joint distribution of the functions evaluated at a finite collection of points.

Since we are only making the case about the marginal distribution of each of the output channels, we can separate them out individually and apply a CLT argument on each one. For any given output channel $g$, the output is a sum $g = \sum_\beta m_\beta h_\beta$ for values $m_\beta$ given by the particular row of $M$. For each coefficient $m_\beta$ that is nonzero (with index $\beta = i$), we can define an element $\tilde{h}_i = m_i h_i$. The output is the sum of such elements $g = \sum_i \tilde{h}_i$.

Now choose any finite collection of points $x_1, x_2, \ldots, x_N$. The random vector $[g(x_1), g(x_2), ..., g(x_N)] = \sum_i [\tilde{h}_i(x_1), \tilde{h}_i(x_2), ..., \tilde{h}_i(x_N)]$. In the limit as the number of channels $c_{in}$ goes to infinity, we can apply the Lyapunov central limit theorem to this sum of random vectors, so long as the number of elements in the sum also goes to infinity. This means that the vector $[g(x_1), g(x_2), ..., g(x_N)]$ converges multivariate Gaussian with mean $\mu$ having entries $\mu_n = \sum_i \mathbb{E}[\tilde{h}_i(x_n)]$ and covariance $\Sigma$ with entries $\Sigma_{nm} = \sum_i \mathbb{E}[\tilde{h}_i(x_n)\tilde{h}_i(x_m)]$. Since we can repeat the same argument for any collection of points $x_1, x_2, \ldots, x_N$, we must deduce that $g$ is a Gaussian process according to the definition. The mean and covariance of $\bar{g}(x) \to \mathcal{GP}(\mu, k)$ follow from the mean and covariance of these finite collections, $\mu(x) = \sum_i \mathbb{E}[\tilde{h}_i(x)] = \sum_i m_i \mathbb{E}[h_i(x)]$ and $k(x, x') = \sum_i \mathbb{E}[\tilde{h}_i(x)\tilde{h}_i(x')] = \sum_i m_i^2 \mathbb{E}[h_i(x)h_i(x')]$.

Conceivably the CLT could be violated during training if a large enough fraction of $M_{\alpha\beta}$ converged to 0 during training, or also if the separate channels became more strongly correlated. We investigate the non-Gaussianity of the features in section 4.7 and find that the features are mostly Gaussian but have somewhat longer tails indicating deviation from the theoretical behavior.

## J   MOMENTS OF RECTIFIED GAUSSIAN RANDOM VARIABLES

Let $\boldsymbol{f} \sim \mathcal{N}(\boldsymbol{\mu}, \boldsymbol{\Sigma})$ be a $d$ dimensional random Gaussian vector. We compute here

$$\mathbb{E}(\text{ReLU}(f_1) \cdots \text{ReLU}(f_d)) = \frac{1}{N_\Sigma} \int_{\boldsymbol{f} > 0} \mathrm{d}^d \boldsymbol{f} \, f_1 \cdots f_d \exp -\tfrac{1}{2}(\boldsymbol{f} - \boldsymbol{\mu})^T \boldsymbol{\Sigma}^{-1}(\boldsymbol{f} - \boldsymbol{\mu}) \tag{33}$$

$$N_{\boldsymbol{\Sigma}} = (2\pi)^{d/2} \det(\boldsymbol{\Sigma})^{1/2} \,. \tag{34}$$

We use the generating function technique. Define

$$Z(\boldsymbol{b}) = \frac{1}{N_\Sigma} \int_{\boldsymbol{f} > 0} \mathrm{d}^d \boldsymbol{f} \exp[-\tfrac{1}{2}(\boldsymbol{f} - \boldsymbol{\mu})^T \boldsymbol{\Sigma}^{-1}(\boldsymbol{f} - \boldsymbol{\mu}) + \boldsymbol{b}^T \boldsymbol{f}] \tag{35}$$

$$= \frac{1}{N_\Sigma} e^{\boldsymbol{b}^T \boldsymbol{\mu}} \int_{\boldsymbol{f} > -\boldsymbol{\mu}} \mathrm{d}^d \boldsymbol{f} \exp[-\tfrac{1}{2} \boldsymbol{f}^T \boldsymbol{\Sigma}^{-1} \boldsymbol{f} + \boldsymbol{b}^T \boldsymbol{f}] \,. \tag{36}$$

Then

$$\mathbb{E}(\text{ReLU}(f_1) \cdots \text{ReLU}(f_d)) = \frac{\partial}{\partial b_1} \cdots \frac{\partial}{\partial b_d} Z(\boldsymbol{b}) \bigg|_{\boldsymbol{b}=0} \,. \tag{37}$$

To compute $Z(\boldsymbol{b})$ we proceed as in Gaussian case. We change variables to

$$\boldsymbol{f} = \boldsymbol{\Sigma} \boldsymbol{b} + \boldsymbol{g} \,, \tag{38}$$

and define $\boldsymbol{z} = \boldsymbol{\mu} + \boldsymbol{\Sigma} \boldsymbol{b}$ to get:

$$Z(\boldsymbol{b}) = e^{\boldsymbol{b}^T \boldsymbol{\mu} + \frac{1}{2} \boldsymbol{b}^T \boldsymbol{\Sigma} \boldsymbol{b}} \frac{1}{N_{\boldsymbol{\Sigma}}} \int_{\boldsymbol{g} < +\boldsymbol{z}} \mathrm{d}^d \boldsymbol{g} \exp[-\tfrac{1}{2} \boldsymbol{g}^T \boldsymbol{\Sigma}^{-1} \boldsymbol{g}] \tag{39}$$

$$= e^{S(\boldsymbol{b})} \Phi^{(d)}(\boldsymbol{z}; 0, \boldsymbol{\Sigma}) \,, \quad S(\boldsymbol{b}) = \boldsymbol{b}^T \boldsymbol{\mu} + \tfrac{1}{2} \boldsymbol{b}^T \boldsymbol{\Sigma} \boldsymbol{b} \,. \tag{40}$$

$\Phi$ being the multivariate standard Normal CDF:

$$\Phi^{(d)}(\boldsymbol{z}; \boldsymbol{\mu}, \boldsymbol{\Sigma}) = \int_{\boldsymbol{g} < +\boldsymbol{z}} \mathrm{d}^d \boldsymbol{g} \psi^{(d)}(\boldsymbol{g}; \boldsymbol{\mu}, \boldsymbol{\Sigma}) \,, \tag{41}$$

$$\psi^{(d)}(\boldsymbol{g}; \boldsymbol{\mu}, \boldsymbol{\Sigma}) = \frac{1}{N_{\boldsymbol{\Sigma}}} \exp[-\tfrac{1}{2}(\boldsymbol{g} - \boldsymbol{\mu})^T \boldsymbol{\Sigma}^{-1}(\boldsymbol{g} - \boldsymbol{\mu})] \,. \tag{42}$$

Now we compute the first two derivatives. Note that in $d = 1$, denoting $\sigma^2 = \Sigma$:

$$\frac{\partial}{\partial z} \Phi^{(1)}(z, 0, \sigma^2) = \psi^{(1)}(z, 0, \sigma^2) \,. \tag{43}$$

In $d = 2$, we can use the conditional probability decomposition to get the required derivatives:

$$\psi^{(2)}(\boldsymbol{g}; 0, \boldsymbol{\Sigma}) = \psi^{(1)}(g_1; \alpha_1 g_2, \beta_1) \cdot \psi^{(1)}(g_2; 0, \Sigma_{22}) \tag{44}$$

$$\alpha_1 = \Sigma_{12} \Sigma_{22}^{-1} \,, \beta_1 = \Sigma_{11} - \Sigma_{12} \Sigma_{22}^{-1} \Sigma_{21} \tag{45}$$

$$\partial_{z_2} \Phi^{(2)}(\boldsymbol{z}, 0, \boldsymbol{\Sigma}) = \psi^{(1)}(z_2; 0, \Sigma_{22}) \int_{-\infty}^{z_1} \mathrm{d}g_1 \psi^{(1)}(g_1; \alpha_1 z_2, \beta_1) \,, \tag{46}$$

$$= \psi^{(1)}(z_2; 0, \Sigma_{22}) \Phi^{(1)}(z_1, \alpha_1 z_2, \beta_1) \tag{47}$$

$$\partial_{z_2}^2 \Phi^{(2)}(\boldsymbol{z}, 0, \boldsymbol{\Sigma}) = -\frac{z_2}{\Sigma_{22}} \psi^{(1)}(z_2; 0, \Sigma_{22}) \Phi^{(1)}(z_1, \alpha_1 z_2, \beta_1) \tag{48}$$

$$+ \psi^{(1)}(z_2; 0, \Sigma_{22}) \partial_{z_2} \Phi^{(1)}(z_1, \alpha_1 z_2, \beta_1) \tag{49}$$

$$\partial_{z_1} \partial_{z_2} \Phi^{(2)}(\boldsymbol{z}, 0, \boldsymbol{\Sigma}) = \psi^{(2)}(\boldsymbol{z}; 0, \boldsymbol{\Sigma}) \,. \tag{50}$$

So denoting $\partial_i = \frac{\partial}{\partial b_i}$, we get:

$$\partial_i Z(\boldsymbol{b}) = (\mu_i + \sum_j \Sigma_{ij} b_j) Z(\boldsymbol{b}) + \underbrace{e^{S(\boldsymbol{b})} \sum_\ell \partial_{z_\ell} \Phi^{(d)}(\boldsymbol{z}, 0, \boldsymbol{\Sigma}) \Sigma_{\ell, i}}_{m_i(\boldsymbol{b})} \tag{51}$$

$$\partial_k \partial_i Z(\boldsymbol{b}) = \Sigma_{ik} Z(\boldsymbol{b}) + (\mu_i + \sum_j \Sigma_{ij} b_j) \partial_k Z(\boldsymbol{b}) \tag{52}$$

$$+ (\mu_k + \sum_j \Sigma_{kj} b_j) m_i(\boldsymbol{b}) + e^{S(\boldsymbol{b})} \sum_{\ell, q} \partial_{z_\ell} \partial_{z_q} \Phi^{(d)}(\boldsymbol{z}, 0, \boldsymbol{\Sigma}) \Sigma_{\ell, i} \Sigma_{q, k} \,. \tag{53}$$

In particular, for the first moment $d = 1$ we have:

$$\mathbb{E}(\text{ReLU}(f)) = \mu \Phi^{(1)}(\mu; 0, \sigma^2) + \psi^{(1)}(\mu, 0, \sigma^2)\sigma^2 ,\tag{54}$$

which coincides with equation 8. Note that $\psi^{(1)}(\mu, 0, \sigma^2) = \frac{1}{\sigma}\psi(\mu/\sigma; 0, 1)$ because of the normalization factor. For the second moments $d = 2$ we have:

$$\mathbb{E}(\text{ReLU}(f_1)\text{ReLU}(f_2)) = \Sigma_{12}\Phi^{(2)}(\boldsymbol{\mu}; 0, \boldsymbol{\Sigma}) + \mu_1\mu_2\Phi^{(2)}(\boldsymbol{\mu}; 0, \boldsymbol{\Sigma}) + \mu_1 m_2(0) + \mu_2 m_1(0)\tag{55}$$

$$+ \sum_{\ell, q = 1, 2} \Sigma_{\ell, 1}\Sigma_{q, 2}\partial_{z_\ell}\partial_{z_q}\Phi^{(2)}(\boldsymbol{z}, 0, \boldsymbol{\Sigma})|_{\boldsymbol{b} = 0} .\tag{56}$$

which can be rewritten in the form of equation 9.

