# OpenReview forum: "Probabilistic Numeric Convolutional Neural Networks"
_ICLR.cc/2021/Conference — ICLR 2021 Poster_

### Official Review · AnonReviewer3 · 2020-10-24
**Promising development in probabilistic convolutional models**

**Rating:** 7
**Confidence:** 3

**Review:**

SUMMARY:

The paper considers the problem of using CNNs on an irregularly sampled grid. It proposes to incorporate the numerical uncertainty related to the disretisation of the domain using an approach inspired by probabilistic numerics. This involves defining continuous convolutional layers as affine transformations of the input GP and use rectified GPs to include the nonlinearity in the mapping. The resulting non-Gaussian distribution is then approximated with another GP in each layer. These are then approximated with a GP with an RBF kernel using Monte Carlo to produce observations with input dependent noise. The model is trained using MAP estimation.


##################################################################

REASON FOR SCORE:

I like the motivation for this work and the paper was generally a pleasure to read. Naturally, many technical challenges arise when trying to approach this problem from a probabilistic perspective; they are discussed in a fair bit of detail and generally seem reasonable. However, I still have a number of questions that are listed below that would help me better understand the main ideas and contributions.


##################################################################

PROS:
+ The motivation for the work seems sound.
+ The main technical contribution seems to be the definition of a continuous convolution on GPs with an RBF kernel. This is presented clearly and in a fair bit of detail.
+ The writing quality is very good. The appendix was very informative as well.


##################################################################

CONS:
- The paper makes quite a few approximations (which mostly make sense) but doesn't give a thorough discussion of the effect of these choices.
- The paper doesn't seem to make full use of the setup (see questions below).
- I found Fig.2, especially 1 and 3, hard to interpret and the figure caption wasn't very informative. It appears from the left figure that the mean is very smooth in all layers. Is that desirable? In the right figure, what is gt on the y-axis?


##################################################################

QUESTIONS and COMMENTS:
- My understanding is that PN approaches are typically used for tasks like, (a) to impose specific priors, (b) to improve the design of the problem or (c) to learn something about the implicit priors in established deterministic methods.
It would be interesting to think about these in the context of your model. For example, (a) is it reasonable to use a GP with a smooth kernel as an interpolant of the inputs? If the kernel choice was not limited by the analytic tractability, would you still choose an RBF kernel? For (b), could you use the model to decide which points in the input domain are the most informative for the end task? I suppose that you would need the full posterior distribution rather than just a point estimate, which would be very tricky given the multi-layer setup. Also, given that your approach seems to outperform existing (deterministic?) models or some tasks, how could you change the existing models to gain a similar advantage?

- It would be interesting to know what is the effect of the various approximations, for example, the Gaussian approximation in 4.6 and the MC in 4.7, on the final task.

- The CLT argument in Appendix I is not very clear to me. The Lyapunov CLT holds in cases of RVs being independent, but not necessarily identically distributed. Is it correct that in your case the RVs are weakly-dependent and identically distributed? What is c in the inequality |c - c'| >> 1? More generally, is the argument in this section some well-known result? It shares some ideas with the argument of Neal [1996] of an infinite limit of NNs being equivalent to GPs but it would be good to either give a more thorough discussion of this argument or provide links to related results in the literature.

- It would be informative to discuss the computational and memory complexity of the proposed approach and other related questions, e.g. the number of samples in the MC steps.

- I'm a little surprised that removing the noise gives worse results in your MNIST experiment. I would guess that this may be more related to some optimisation phenomenon than to actual model. For example, if instead of using the data-dependent noise parameter as described in Sec. 4.7 you used a spherical Gaussian noise with fixed sigma for all inputs, would that give better results than discarding the noise entirely? And how would it compare with your data-dependent noise? Using spherical Gaussian noise relates with some of the practices in DGP models where the input to each layer is assumed to be a noisy version of the outputs of the previous layer.

- Related literature: It might be interesting to compare to some DGP models that also use convolutions, e.g. Deep Gaussian Processes with Convolutional Kernels [2018] by Kumar et al and Deep convolutional Gaussian processes [2019] by Blomqvist et al.

##################################################################

MINOR:

- p. 1 - Probalistic Numeric
- p. 6 - the number the number
- p. 15 - infinitesmal
- p. 18 - many many observations
- p. 19 - a variant a variant

---

> ### Author Response · Authors · 2020-11-24
> **Response**
>
> Thank you for your detailed and thoughtful review.
>
> We’d like to correct a possible misunderstanding in the workings of the method. We don’t use Monte Carlo sampling of the previous layer to determine the next RBF GP. Rather, we feed the mean and uncertainty into the RBF GP directly as observations, where the uncertainty forms the noise model determining the output uncertainty and modulating the weighting of the mean values. The model that we term PNCNN w/o sigma removes the uncertainty not just here, but also reducing the probabilistic ReLU to a standard one.
>
> **a)** Regarding the expanded uses of Probabilistic Numerics and specific priors, for problems like Superpixel MNIST and PhysioNET the RBF kernel prior is quite sensible however for other problems with less continuity it would be important to use other kernels like Matern. We hope to see the extension to other tractable kernels and PDE/convolution operators pairs in future work. Even without that, other kernels like a sum over RBFs with different lengthscales could be implemented trivially.
>
> **b)** Using most informative input points: this is something we had experimented with. In the current setting we optimized the locations of the evaluation points at the intermediate layers of the network with respect to the marginal likelihood of the GPs. Since we have an approximation of the full distribution and not a point estimate, we can do this directly and without needing the labels. However, we didn't feel that the marginal improvements for doing so warranted the additional complexity of the method as it also requires performing this optimization for each test input.
>
> **c)** Outperforming deterministic models and abstracting the relevant factors: probabilistic numerics are primarily applied to linear problems: e.g. integration, matrix inversion, solution to linear PDE, as a result the output mean and uncertainty can be decoupled in these problems. In these situations the mean corresponds exactly to or very nearly to the output of a deterministic algorithm, and the uncertainty is an ancillary output that does not directly affect the mean predictions. However, because PN in PNCNN are applied to a fundamentally nonlinear operation, the mean and uncertainty interact, yielding output mean predictions that are distinct from a deterministic model. This is why the PNCNN model with uncertainty performs much better than the PNCNN model without uncertainty. We see this as an underexplored benefit of probabilistic numerics.
>
> We hope this clears up confusion about why removing uncertainty harms performance. The reason is _not_ just an optimization phenomenon, but is precisely the reason for using probabilistic numerics in this context. Accounting for the distribution of different functions consistent with the (sparse) observations yields better predictions of the continuous network on the unknown continuous input than a non probabilistic approximation.
>
> **Central Limit theorem:** We agree that the statement of the CLT in the text could be made clearer and more precise. c here is the channel index and the requirements are the weak dependence, but not necessarily that the channels are identically distributed. We will add additional discussion and citations to relate back to the Bayesian infinite limit of NNs even though the argument being made here is distinct.
>
> **Effect of Approximations:** The two major approximations that we make are the above mentioned central limit theorem (which only holds approximately) and the projection of the approximately gaussian (but not RBF) process from the CLT to an RBF GP. We describe limitations of this second approximation in appendix H, but will make sure that this is more clearly referenced in the text. We also evaluate these approximations empirically by figure 2 right where the extent of non gaussianity is quantified with the QQ plot for each layer. These curves show that these two approximations yield heavier tails than a gaussian would have in the later layers of the network. The ‘gt’ prediction here is from the highest resolution model using 256 pixels. The plot captures how the predicted spatial uncertainties due to sparse spatial data are calibrated to the actual values when more spatial data (pixels) are observed. Thank you for pointing this out and we will reword this caption to make this clearer.
>
> The total computational complexity is O(n^3LD) where n is the number of observations per data point, D is the number of data points, and L is the number of layers and O(n^2L) memory complexity. This is in contrast to GP methods that use O(D^3) compute and O(D^2) memory.
>
> While the two convolutional GP papers you mention are significant, they work only for gridded data and cannot be applied to irregularly spaced data without substantial changes to the methodology. One of these adaptions is graph convolutional gaussian processes (GCGP) which we _do_ compare to and outperform on Superpixel MNIST (4.2% error vs 1.24%).

---

### Official Review · AnonReviewer2 · 2020-10-28

**Rating:** 6
**Confidence:** 4

**Review:**

Summary: This work presents an uncertainty aware continuous convolutional layers for learning from continuous signals like time series/images. This work is most useful in the setting of irregularly sampled data. Gaussian processes (GP) are used to represent the irregularly sampled input. The proposed continuous convolutional layers can be directly applied to input Gaussian Process in a closed form, which subsequently outputs another GP with transformed mean and variance. Finally, the continuous convolutional layers are parameterized in terms of the flow of a PDE. The use of GP for feature representation and the ability of continuous convolutional layers to take GP as input provides the model with the ability to propagate uncertainty.

Positives:
1. The problem addressed is very important in many domains.
2. The proposed approach is novel for learning from irregularly sampled data.
3. Empirical results show that proposed model performs well.

Concerns:
1. Given the time complexity of Gaussian processes, could the authors comment on the run-time of the proposed method as compared to standard CNN?
2. This work is missing comparisons with one of the recent work focusing on the same problem (Rubanova et al, 2019). It would be interesting to also compare the run-time of both these methods.
3. In Section 2 para 3, the authors mention that Gaussian process adapter (Li and Marlin, 2016) operates on a deterministic signal
on a regular grid and cannot reason probabilistically about discretization errors. I don't think this is true. Infact, GP adapter is an uncertainty aware classification framework which has the ability to pass the uncertainty through to classification framework similar to the proposed approach. This makes these two approach more similar in what they are trying to achieve and I would expect the authors to compare the performance of these methods.
5. The authors repeatedly discuss that their approach does not require any discretization unlike some the previous approaches based on GP(mentioned in Section 2). But in Section 4.7, they mention that their approach requires to evaluate Gaussian Process at a set of points which is similar to what was done in Li and Marlin(2016). Could the authors clarify this?

Additional Comments:
1. Could the authors comment on the possibility of using the proposed approach for interpolation in irregularly sampled time series?

I would be happy to increase the score once my concerns are addressed.

---

> ### Author Response · Authors · 2020-11-24
> **Response**
>
> We thank you for your detailed and attentive review. Below we address your questions and concerns.
>
> **1) Runtime:**
> For the scale of inputs we evaluate in the paper (with 75-256 sampling locations), the runtime of PNCNNs are similar to other methods that can work with irregularly spaced data. As mentioned in appendix F, both models take ~2 hours to train. For the time series, the Latent ODE (Rubanova et al, 2019) would take ~10 hours for the same number of epochs (as reported in Horn et al, 2019).
>
>  The naive GP inference scales O(n^3) with the number of sampling locations, so we would expect runtimes to increase more sharply for larger inputs for PNCNN than for the other methods. We leave the integration of the scalable GP methods to future works. If restricted to a regular square grid image, and compared with a standard CNN on a problem of comparable size, a standard CNN would train in minutes; however, this is of course not the target application of the approach and the standard CNN cannot be applied to irregularly spaced data. Moreover, the kinds of inputs where discretization and sampling errors are especially important are precisely those with a small number of spatial samples, therefore mitigating the computational burden.
>
> **2) Comparison to Latent ODEs:**
> We cite the approach of Rubanova et al, 2019, but did not include the result due to the discrepancy in the training regime between Rubanova et al, 2019 and Horn et al, 2019. Rubanova et al perform semi supervised learning where only ⅓ of the data is labeled whereas we adopt the setting from Horn et al where all of the training data is labeled. However, Horn et al have since updated their paper with an additional comparison to Latent ODEs in their setup which gets 85.7 AUROC and 50.7 AP (which is slightly worse than PNCNN), however they mentioned that due to the much longer training time they did not tune this model as much as the baselines. We will reference this result for the table in the final version.
>
> **3) GP adapter:** The comment we made here was aimed at the fact that the GP adapter is a method split into two parts: a GP that is used to probabilistically generate completions of the input evaluated on a regular grid, and a separate non probabilistic CNN classifier (they also considered an MLP and logistic regression which were not as performant). This separate non probabilistic CNN classifier only works with a single imputation at a time that is on a discretized grid. Regardless of the imputations of the GP, this CNN will carry a discretization error that is not explicitly modeled. Given a fixed grid, passing multiple imputations onto the grid through the CNN will not capture the uncertainty due to the missing values between the grid points.
>
>  Whereas the GP adapter can describe uncertainty by taking many sample completions of the input and feed these samples through the discrete and deterministic classifier, our PNCNNs differ in two regards. PNCNNs define an _intrinsically continuous operator_, and we derive how that operator propagates the _uncertainty distribution_ through the layers and not merely samples from the distribution.
>
> **4)** The continuous conv layers can be applied to the GP feature map in closed form and without discretization. In the projection to RBF GP step, we evaluated the transformed GP at a discrete set of locations (that are furthermore not on a grid), and those evaluations reflect the uncertainty of computing the convolved input depending on the function over the domain and not just at the evaluation locations. This is in contrast to a discrete convolutional model that only aggregates over the prefixed grid points. In the GP adapter, it is not the evaluation of the GP which is discretized, but rather the convolutional layers of the CNN following the GP.
>
> Additional comment **1) Interpolation of irregularly sampled time series:**
> Unlike GPs, PNCNNs are fundamentally not trying to solve this problem. A perhaps better way of conceptualizing our method is as a network that takes as input a GP and outputs another GP or a vector of features.
>
> We appreciate the reviewers comments and hope that the separation from prior work is now clearer.

---

### Official Review · AnonReviewer4 · 2020-10-28
**Probabilistic numerics inspired DNN using GPs**

**Rating:** 7
**Confidence:** 1

**Review:**

This paper is way out of my comfort zone so I can only provide very general high level comments.

First, the motivation for the paper seems very clear. For deep learning to work, we often have to discretize, and have to do so at lower resolution, which induces errors we typically ignore.

Second, the architecture is well described and illustrated.

Some references (e.g. p1 Cockayne 2019) are in the wrong format.

I didn't check the math in all details, but the three experiments with superpixel resolution, generalization to new resultions and and irregular time series appear convincing.

---

> ### Author Response · Authors · 2020-11-24
> **Response**
>
> We appreciate the review. While the paper is somewhat technical, we hope to improve the readability with more familiar notation in certain places and adding additional definitions as suggested by another reviewer. Thank you for pointing out the reference format errors, we will fix these.

---

### Official Review · AnonReviewer1 · 2020-10-28
**Good, enjoyable paper; flimsy CLT; hard to follow**

**Rating:** 7
**Confidence:** 3

**Review:**

# Summary

To work with data that is not sampled on a grid, this work represents the activations of a neural network using a Gaussian process with RBF covariance. Convolutions can be applied in closed form, but nonlinearities get projected to another RBF-posterior GP by

1) calculating the first two moments of the post-nonlinearity activation, and
2) projecting the moment-matched GP to a finite set of noisy (independent) observations,

which can then be fitted with an RBF GP again, and so on until the end. The proposed method works very well for irregularly-sampled MNIST, and fairly well for the PhysioNet2012 data set for ICU mortality.

# Comments

The paper references related work, makes a meaningful contribution, and I think the empirical methodology is sound. Thus, I think it should be accepted. As negatives, there are a few small issues with the paper, and it is hard to follow.

## Central limit theorem

The CLT argument referenced in section 4.6, only works if all the elements of M are equal. (As a counterexample: suppose only one column of M is 1, and all the other ones are 0. Then, if the corresponding input process is non-Gaussian, all the outputs will be non-Gaussian).

In fact, the argument in the appendix forgoes coefficients for every $g_c$ altogether. It is correct, but it is not the same object in section 4.6.

I don't buy your CLT argument, but I think the rest of the paper is still great without it. Simply, you're doing a moment-matching approximation, like Expectation Propagation (Minka, 2001b).

I would consider dropping the CLT claims. Or, you could argue that in practice the elements of M are similar.

You could also argue that the elements of M are close to independent with mean zero. If M is also fairly large, you can argue that $f$ is close to a GP, following the theoretical arguments from the infinitely wide NN literature (Matthews et al., 2018; or generalizations thereof such as Yang, 2019).

Minka, T. P. (2001). Expectation propagation for approximate Bayesian inference. UAI (pp. 362-369)
Matthews, A G de G, et al. (2018). Gaussian process behaviour in wide deep networks. https://arxiv.org/abs/1804.11271
Yang, G., (2019). http://papers.neurips.cc/paper/9186-wide-feedforward-or-recurrent-neural-networks-of-any-architecture-are-gaussian-processes

## Some experiment and figure clarification

In section 4.7 you talk about a collection of points ${x_i}_{i=1}^N$ to which you project the activation. Are these points the same for every layer? Are they always the points at which the network is currently being evaluated (I understand these are points for one input point, i.e. sampling locations for an MNIST image)?

For figure 2, how many test images do you use? Is it one, with varying number of sampling locations per image?

For Figure 3, how do you change the test-time resolution of a normal CNN, one that has discretely specified pixels? Do you linearly interpolate their values? This should be explained.

## Other clarification requests and suggestions

I found the paper hard to follow, and some things are unclear. This is understandable, because it uses mathematical tools unusual for an ML paper, but you can still make it easier for the reader. The rest of my review is some requests for clarity and writing suggestions.

*Taylor expansion above eq. 3:* it is an expansion around $x$ of $g(x + a)$, no? I believe stating this simple fact after "Verify by Taylor expansion [around $x$ of $g(x + a)$] would make it much less confusing.

What is the "[a, b]" operator at the top of page 5?

*Eq 9*:  I think $\nabla\nabla^T {\boldsymbol \Phi}({\boldsymbol \mu}, {\boldsymbol \Sigma})$ is supposed to represent the Hessian of ${\boldsymbol \Phi}({\boldsymbol \mu}, {\boldsymbol \Sigma})$, but I'm not sure. It's not quite standard notation and should be mentioned in the paragraph above.

In section 4.7, when projecting the current activations to a set of points, you lose the covariance. It's good that you comment on this in Appendix H, and it would be better if you pointed the reader to it.

I think specifying that you use Adam in section 4.8 is helpful to the reader and doesn't take much space.

Appendix F: $3 \cdot 10^{-3}$, not $310^{-3}$.

Are you going to release the code?

---

> ### Author Response · Authors · 2020-11-24
> **Response**
>
> Thank you for your detailed and constructive feedback.
>
> Regarding the central limit theorem, you bring up the point that the argument only works if all of the elements of $M$ are equal, with the clear counterexample when many elements are zero. However, in fact the CLT also applies when the values of $M$ are distinct so long as they are nonzero.
> Explicitly, consider$f = \sum_{j=1}^N m_j h_j$, where $h_j$ are random variables (here m represents a specific row of M for a given output channel). So long as none (or more generally, a vanishing fraction as N goes to infinity) of the $m_j$'s are not 0, we can safely redefine $\tilde{h}_j = m_jh_j$ and apply the CLT to the transformed variables $f = \sum_j \tilde{h}_j$. However, we agree that this point was not made in the paper, and that we shall improve on the clarity of the derivation, discussing the assumptions more clearly.
>
> However to your point, one could imagine elements of $M_{ij}$ converging to 0 during training, or violations of the weak independence property yielding deviations which are not entirely characterized by the CLT. We will add these caveats to the discussion.
>
> As you point out we might be able to make the argument used in the GP limit where $M$ is random, however this goes counter to the kind of uncertainty we are representing which is not Bayesian about the weights but about the inputs. Since we are not considering different draws of the weights $M$ from a prior or posterior, but instead fixed learned values of the weights and different completions of the input we can’t make this argument. It is an interesting point of relation though, and we will comment on this in Appendix I and add the relevant citation.
>
> Clarification:
> Projection points $\\{x_i\\}$: yes these are the same for each layer. For the image experiments, these are set to be the same as the input sampling locations. However, they can be changed and we briefly experimented with having these be learned locations before reverting to the simpler placement for the paper.
> Figure 2 was based on one minibatch of size 50 from the test set, but closely reflects the results from the whole test set.
> For the experiment in Figure 3 (unlike elsewhere in the paper) we used the standard version of MNIST so as to be able to compare to the standard CNN. Here the tested sampling locations are all regular square grids of size 4x4, 6x6, 8x8, …, 16x16 so they can be fed into the CNN for a fair comparison.
>
> $[a,b]$ operator: this is the commutator $[a,b] = ab-ba$ measuring the extent to which the operators a and b fail to commute, and $\nabla\nabla^T\Phi(\mu,\Sigma)$ is indeed the Hessian. We’ll mention this in the text.
>
> Readability: We agree with the points raised and will fix them to improve readability in the final version.
> Code release: We plan on releasing the code on acceptance.

---

> > ### Comment · AnonReviewer1 · 2020-11-25
> > **Happy with proposed amendments and arguments -- please make sure you actually amend!**
> >
> > I'm broadly happy with your reply, and also that the other reviewers mostly recommend accepting this paper as well.
> >
> > > one could imagine elements of $M_{ij}$ converging to 0 during training
> >
> > In my experience, this happens quite a bit if one uses weight decay (L2 regularization) during training, so it is important.
> >
> > >  We will add these caveats to the discussion.
> >
> > Good, please do. You don't have to add citations for the GP limits, if you don't think they are relevant -- but it's something that would convince some readers.
> >
> > I think citing Minka and mentioning that your layer-wise approximation matches moments like EP (if you agree with me on this) is somewhat more important.
> >
> > Please also make sure you include the info from the "Clarification:" paragraph somewhere in the paper or the appendices.
> >
> > Thank you for your work!

---

### Decision · Program_Chairs · 2021-01-07
**Final Decision**

**Decision:**

Accept (Poster)

**Comment:**

This is a fairly technical paper bridging deep learning with uncertainty propagation in computations (i.e. probabilistic numerics). It is well structured, but it could benefit from further improvements in readability given that there are only very few researchers that are experts in all sub-domains associated with this work. Given the above, as well as low overall confidence by the reviewers, I attempted a more thorough reading of the paper (even if not an expert myself), and I was also happy to see that the discussion clarified important points. Overall, the idea is novel, convincing and seems well executed, with good results. The technical advancements needed to make the idea work are fairly complicated and are appreciated as contributions, because they are expected to be useful in other applications too (beyond irregular sampled data) where uncertainty propagation matters.